# Structural basis of Gip1 for cytosolic sequestration of G protein in wide-range chemotaxis

Takero Miyagawa[1,2], Hiroyasu Koteishi [2,4], Yoichiro Kamimura[2], Yukihiro Miyanaga[1], Kohei Takeshita[3,5], Atsushi Nakagawa [3] & Masahiro Ueda [1,2]

G protein interacting protein 1 (Gip1) binds and sequesters heterotrimeric G proteins in the cytosolic pool, thus regulating G protein-coupled receptor (GPCR) signalling for eukaryotic chemotaxis. Here, we report the underlying structural basis of Gip1 function. The crystal structure reveals that the region of Gip1 that binds to the G protein has a cylinder-like fold with a central hydrophobic cavity composed of six α-helices. Mutagenesis and biochemical analyses indicate that the hydrophobic cavity and the hydrogen bond network at the entrance of the cavity are essential for complex formation with the geranylgeranyl modification on the Gγ subunit. Mutations of the cavity impair G protein sequestration and translocation to the membrane from the cytosol upon receptor stimulation, leading to defects in chemotaxis at higher chemoattractant concentrations. These results demonstrate that the Gip1-dependent regulation of G protein shuttling ensures wide-range gradient sensing in eukaryotic chemotaxis.

[1] Graduate School of Frontier Biosciences, Osaka University, Suita, Osaka 565-0871, Japan. [2] Center for Biosystems Dynamics Research (BDR), RIKEN, Suita, Osaka 565-0874, Japan. [3] Institute for Protein Research, Osaka University, Suita 565-0871, Japan. [4]Present address: Department of Life Science, Graduate School of Life Science, University of Hyogo, 3-2-1 Kouto, Kamigori-cho, Ako-gun, Hyogo 678-1297, Japan. [5]Present address: Advanced Photon Technology Division, RIKEN SPring-8 Center, 1-1-1 Kouto, Sayo-cho, Sayo-gun, Hyogo 679-5148, Japan. Correspondence and requests for materials should be addressed to Y.K. (email: ykamimur@riken.jp) or to M.U. (email: masahiroueda@fbs.osaka-u.ac.jp)

Heterotrimeric G proteins (G proteins) play a pivotal role in G protein-coupled receptor (GPCR) signalling in the detection of various environmental stimuli, including hormones, neurotransmitters, light, odourants, and chemoattractants[1–3]. G proteins consist of Gα and tightly bound Gβγ subunits. Gα is a guanine nucleotide-binding protein with intrinsic GTPase activity, and its GDP-bound form can complex with Gβγ subunits, resulting in an inactive state. G proteins are activated by ligand-bound GPCR, which behaves as a guanine nucleotide exchange factor (GEF) to catalyse GDP–GTP exchange at the Gα subunit. The GTP-bound Gα subunit dissociates from the Gβγ subunits and achieves signal transduction by interacting with effectors until the bound GTP is hydrolysed to GDP by GTPase-activating proteins (GAPs), such as regulatory G protein signalling (RGS) proteins[4–6]. The Gβγ subunits also serve as signal transducers to downstream pathways through different effectors[7,8]. These reactions occur on the plasma membrane, as ensured by lipid modifications at the N terminus of the Gα subunit and the C terminus of the Gγ subunit[9]. The structural basis of GPCR signalling has been extensively studied to reveal the molecular function of each signalling component, as reviewed in refs. [10–13].

Eukaryotic chemotaxis is widely observed in development, wound healing, and immune response[14,15]. G protein signalling enables the directional migration of chemotactic cells, including mammalian neutrophils and the social amoeba *Dictyostelium discoideum*, towards the chemoattractant source over broad concentration ranges. *Dictyostelium* cells show chemotaxis towards cyclic adenosine monophosphate (cAMP) via its GPCR, cAR1, and cognate G proteins, such as Gα2Gβγ, whose activation is transduced to multiple signalling pathways[16]. The wide-range chemotaxis involves the desensitization of GPCR cAR1 through its phosphorylation[17] and adaptation downstream of G proteins. In fact, sustained Ras activation by the genetic deletion of Ras negative regulators, NfaA or C2GAP1, impaired the wide-range chemotaxis[18,19]. In addition to these mechanisms, a recent study revealed another mechanism at the G protein level for wide-range chemotaxis[20]. Heterotrimeric G proteins are fully activated at relatively low cAMP concentrations[21], but cells still show chemotactic ability at higher concentration ranges[22]. Along with its regulation of the nucleotide form, recent reports have found that G protein interacting protein 1 (Gip1) regulates G protein signalling for wide-range chemotaxis[20]. Cytosolic Gip1 forms a complex with G proteins, and a portion of G proteins are sequestered in cytosolic pools and prevented from localizing on the membrane, in which Gip1 prefers binding with the heterotrimeric form of G proteins mainly through the βγ subunit[20]. Gα2Gβγ on the membrane mediates chemotactic signalling upon receptor stimulation under chemoattractant gradients[16,23,24]. The cytosolic pool also plays an essential role in chemotactic signalling[20]. Chemoattractant stimulations induce the translocation of cytosolic G proteins to the membrane[20,25], which is likely to supply more G proteins for receptor-mediated chemotactic signalling at higher concentration ranges. This reaction reinforces the redistribution of G proteins on the membrane along the chemical gradients. In fact, a loss of the cytosolic pool in Gip1-deficient cells impairs chemotaxis at higher concentration ranges but not at lower ones. Therefore, Gip1 is a regulator of G protein shuttling between the membrane and the cytosol for wide-range chemotaxis.

Gip1 consists of two regions, an N-terminal Pleckstrin-homology (PH) domain and a C-terminal region. While the PH domain provides the impetus for G protein membrane translocation from the cytosol in a cAMP-dependent manner, the C terminus of Gip1 is sufficient for binding and sequestering G proteins in the cytosol and has weak homology with tumour

necrosis factor α-induced protein 8 (TNFAIP8) family proteins[20,26]. The mammalian TNFAIP8 family proteins include TNFAIP8 and TNFAIP8-like 1–3 (TIPE1–3) proteins; the structures of these proteins have been solved, except for that of TIPE1, and they share a common configuration that includes a hydrophobic cavity[27–29]. These proteins regulate immunity and tumorigenesis[26], and TIPE3 has been reported to show lipid transfer activity through the interaction between its hydrophobic cavity and lipids[28]. Despite these studies, the molecular mechanism of Gip1–G protein complex formation remains unknown.

In this study, we reveal the molecular basis of Gip1-dependent G protein sequestration. The X-ray crystal structure of the C-terminal region of Gip1 shows cylinder-like folding with a central hydrophobic cavity. Mutagenesis and biochemical analyses indicate that the hydrophobic cavity and the hydrogen bond network at the entrance of the cavity are essential for complex formation with the geranylgeranyl modification on the Gγ subunit. Mutations of the cavity impair G protein sequestration and translocation to the membrane from the cytosol upon receptor stimulation, leading to defects in chemotaxis at high concentrations. These results demonstrate that the Gip1-dependent regulation of G protein shuttling ensures wide-range gradient sensing in eukaryotic chemotaxis.

## Results

**Structural determination of the C-terminal region of Gip1.** To gain mechanistic insights into the complex formation that enables the cytosolic sequestration of G proteins, we conducted a structural analysis of Gip1. We solved the crystal structure of the G protein-binding region of Gip1, Gip1(146–310), at 1.95-Å resolution (PDB 5Z1N) (Supplementary Table 1). Gip1(146–310) consisted of seven α-helices designated α0 to α6 from the N terminus (Fig. 1a, Supplementary Figure 1a–c). Among them, helices α1 to α6 formed a cylinder-like structure with a centrally located hydrophobic cavity 22 Å in depth and 10 Å in diameter. We noticed that there was a tetrahedral area of electron density at the entrance of the cavity that extended to two long regions of electron (Supplementary Figure 2a, b). To demonstrate interaction with an exogenous molecule, lipids were extracted from bacteria-purified Gip1(146–310) and analysed by thin layer chromatography. These regions of density were confirmed as a mixture of phosphatidylethanolamine (PE) and phosphatidylglycerol (PG), two major bacterial glycerophospholipids[30] (Supplementary Figure 2c). The lipid was surrounded by one water molecule and 22 residues that mostly formed hydrophobic interactions (Fig. 1b, Supplementary Movies 1 and 2). In contrast to the hydrophobic interior, the surface of Gip1 was relatively charged without any notable hydrophobic regions (Fig. 1c). The entrance of the cavity was negatively charged, and the side showed positively charged patches.

In addition to the 1.95-Å resolution structure, we obtained another crystal structure of Gip1(146–310) at 2.74-Å resolution (PDB 5Z39) (Fig. 2a, Supplementary Figure 3a, Supplementary Table 1). The two structures at 1.95 and 2.74 Å resolution were designated Form I and Form II. Superimposition of the Cα atoms of Form I and Form II revealed that the r.m.s. deviations between helices α2 to α5 are relatively low, while the intervening loop regions of each α-helix have slightly higher r.m.s. deviations, suggesting structural flexibility (Supplementary Figure 3b). The loop region between helices α5 and α6 (a.a. 286–291) has high r.m.s. deviations due to the kink in the α6 helix of Form I, which probably exists because of penetration by water molecules (Supplementary Figure 3c). Furthermore, the high r.m.s. deviations of helices α1 and α6 helices were derived from rotational

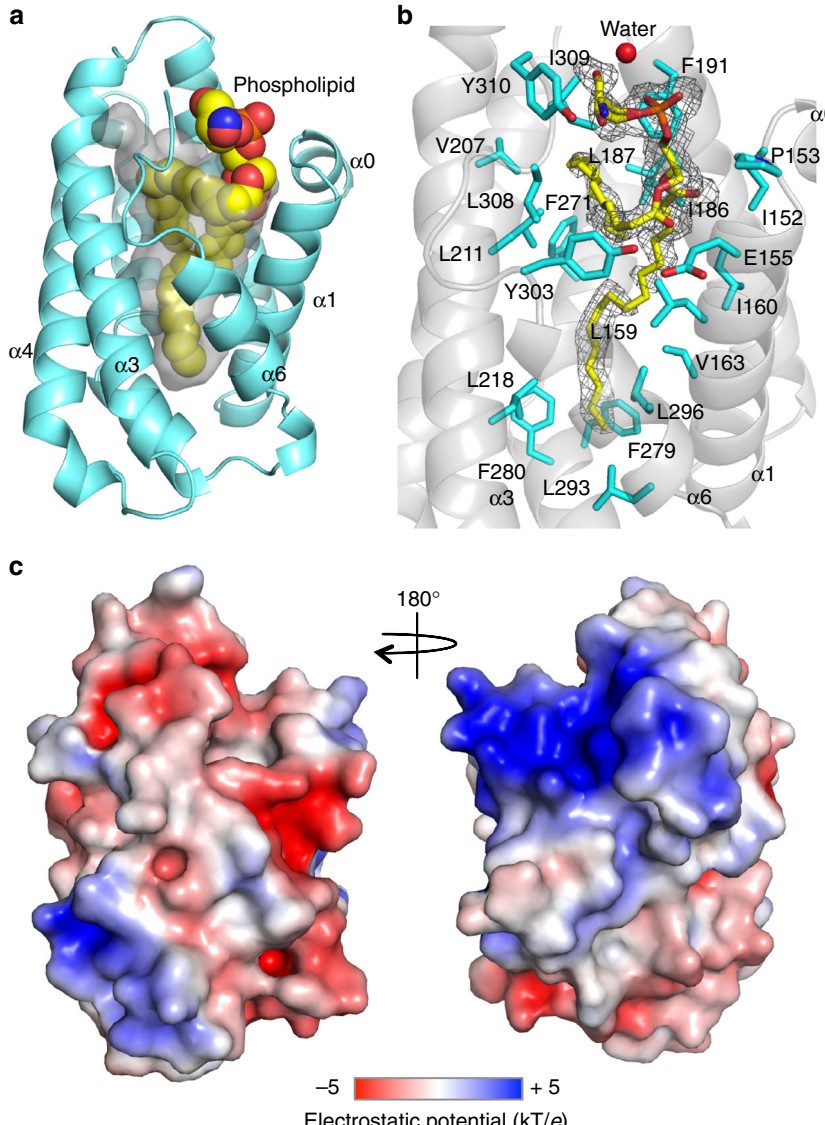

**Fig. 1** Crystal structure of the G protein-binding domain of Gip1. **a** Overall structure of Gip1(146–310) and a phospholipid represented by a cartoon and ball model. The surface of the cavity is shown in grey. **b** Residues and water within van der Waals distance from the phospholipid. The phospholipid is shown with the 2mFo-DFc electron density map contoured at 1.0$\sigma$. The water is shown as a red ball. **c** Electrostatic potential at the molecular surface of Gip1 (146–310), ranging from blue (+5 kT/$e$) to red (−5 kT/$e$)

movements by both helices that result in different hydrogen bonds (Fig. 2b, c). For example, the hydrogen bonding partners of aspartic acid, the 208th amino acid residue (Asp208), and Arg212 were Leu308 and Thr304 in Form I but Tyr303 in Form II (Fig. 2c). The hydrogen bond pattern changes are associated with different configurations of the C terminus of the α6 helix, including a directional change of the side chain at Glu307 (discussed later) (Supplementary Figure 3d). This structural difference resulted in a slight change in the cavity shape and entrance (Fig 2a–c, Supplementary Figure 3e–g). Accordingly, a lipid inside the cavity is in different locations in Form I and Form II (Supplementary Figure 3h).

Although there are differences in the protein sequences of Gip1 and TNFAIP8 family proteins (Supplementary Figure 4), the overall structure of Gip1(146–310) was similar to that of several TNFAIP8 family proteins, including TNFAIP8 (PDB 5JXD), TIPE2 (PDB 3F4M) and TIPE3 (PDB 4Q9V), with r.m.s. deviations of 2.0, 2.7 and 2.1 Å for 88 amino acids (a.a. 173–196, 205–221, 236–256, 261–286 from Gip1 Form I),

respectively[27–29]. Overall, these data suggest that the cylinder-like fold with a hydrophobic cavity is a structural hallmark of TNFAIP8 family proteins and is required for their lipid interactions. Despite this structural similarity, there were some differences between Gip1 and TNFAIP8 family proteins. First, the C-terminal tail of Gip1 was positioned along the α3 helix through hydrogen bond formation, whereas the tails of TNFAIP8, TIPE2 and TIPE3 were directed towards the cavity (Fig. 2d). Second, the direction of the α0 helix differed between Gip1 and TIPE3 due to hydrogen bonding interactions (Fig. 2d).

**Prenyl modification on Gγ is essential for Gip1 interactions.** Although it is unknown whether the bound phospholipid has physiological significance, the structural results suggested that Gip1 accommodates lipid modifications of G proteins inside the hydrophobic cavity during binding. To test this hypothesis, we examined the involvement of lipid modifications of G proteins complexed with Gip1. First, we confirmed lipid modifications on

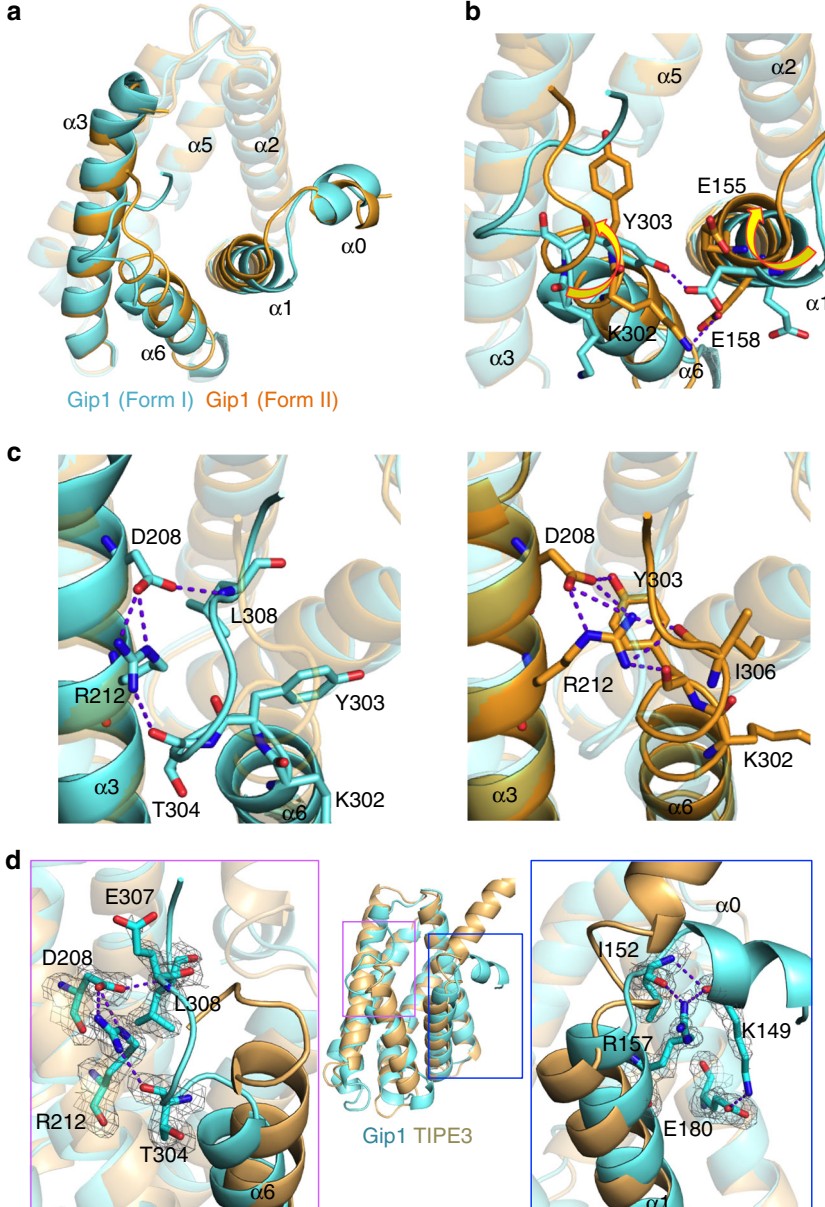

**Fig. 2** Structural comparison of two forms of Gip1 and of Gip1 and human TIPE3. **a** Comparison of two crystal structures of Gip1(146–310). Two forms of Gip1 (Form I in cyan and Form II in light orange) are superimposed. **b** Rotational movement at α1 and α6. The movement reconstructs a hydrogen bond between α1 and α6. Hydrogen bonds are shown as dashed lines. **c** Reconstruction of the hydrogen bonding network at α3 and α6. Structural change induces the reconstruction of the hydrogen bonding network, including Asp208 and Arg212. Hydrogen bonds are shown as dashed lines. **d** Structural comparison of Gip1(146–310) in cyan and TIPE3 (PDB 4Q9V) in pale orange by superimposition. Hydrogen-bonded residues of Gip1 are shown as stick models with the 2mFo-DFc electron density map contoured at 1.0σ. Hydrogen bonds are shown as dashed lines

the Gα2 and Gγ subunits. As expected from the primary amino acid sequences[31], mass spectrometric analysis showed that the Gγ and Gα2 subunits were subject to the geranylgeranylation of Cys66 and the myristoylation of Gly2, respectively (Fig. 3a, Supplementary Figure 5).

Since Gip1 binds mainly to Gβγ[20], we focused on the geranylgeranylation of Gγ and created a mutant lacking this modification by deleting the CAAX (C, A, and X indicate cysteine, aliphatic, and any amino acid residue, respectively) motif at the Gγ C terminus[32]. Gγ(ΔCAAX) was able to form a complex with Gβ, although its localization was mostly in the cytosol, in contrast to wild-type Gβγ, which was observed on the membrane (Fig. 3b, c). Additionally, Gγ(ΔCAAX) could not rescue the developmental defects of gγΔ cells, showing that the

lipid modifications are indispensable for Gγ function (Supplementary Figure 6a). Mutant G proteins containing Gγ(ΔCAAX) were not able to bind with Gip1 (Fig. 3c). To more directly show the importance of this lipid modification on Gγ, we carried out an in vitro-binding assay between Gip1 and Gβγ[20]. Gβγ subunits without Gα were mixed with bacterially purified Gip1. Gip1 bound to wild-type Gβγ but not to Gβγ(ΔCAAX) (Fig. 3d). Furthermore, the complex formation was competitively inhibited by the addition of geranylgeranyl pyrophosphate (Fig. 3e). In this experimental setting, 2 nmol of geranylgeranyl pyrophosphates at 1% showed an effect on 4 pmol of Gip1 (Supplementary Figure 6b). We also examined whether other lipids, including farnesyl pyrophosphate and myristic acid, influenced complex formation. Farnesyl pyrophosphate showed slight impairment of the Gip1–G

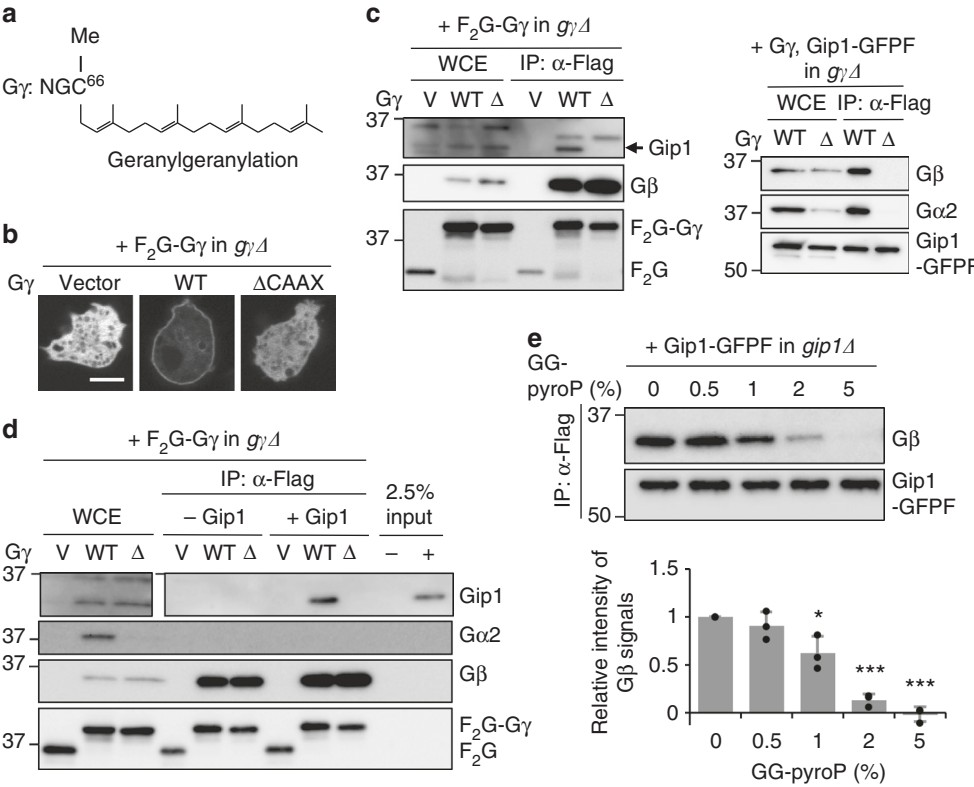

**Fig. 3** Recognition of the prenyl group on Gγ by the hydrophobic cavity of Gip1. **a** Schematic diagram of Gγ modification. C-terminal cysteine (C[66]) is geranylgeranylated and methylated (Me). **b** Subcellular localization of Gγ in a living cell. Flag-Flag-GFP (F$_2$G) tag alone (vector) or F$_2$G-tagged Gγ(WT) or Gγ(ΔCAAX) in *gγΔ* cells. Scale bar, 5 μm. **c** Co-immunoprecipitation of Gγ or Gip1. F$_2$G-Gγ(WT) or F$_2$G-Gγ(ΔCAAX) was expressed in *gγΔ* cells (left). GFP-Flag-tagged Gip1 (Gip1-GFPF) was coexpressed with Gγ(WT) or Gγ(ΔCAAX) in *gγΔ* cells (right). Pull-down samples of Gγ (left) and Gip1 (right) were immunoblotted with the indicated antibodies. **d** In vitro interaction between Gβγ and purified Gip1. Gβγ subunits were bound to beads and incubated with purified full-length Gip1. **e** Competitive dissociation of G proteins from Gip1 by geranylgeranyl pyrophosphate (GG-pyroP). The data were normalized relative to the band intensities without GG-pyroP and presented as the mean ± SD of three independent experiments (*n* = 3, *P < 0.05, ***P < 0.001 versus 0% GG-pyroP, two-tailed unpaired Student's *t*-test)

protein complex but myristic acid did not (Supplementary Figure 6c). This evidence shows that geranylgeranyl modification on the Gγ subunit is essential for interaction with both Gip1 and the plasma membrane.

We previously showed that Gip1 sequesters G proteins in the cytosol[20]. Consistently, wild-type Gβγ but not Gβγ(ΔCAAX) bound to Gip1 only in the cytosolic fraction (Fig. 4a). To determine the physiological relevance to complex formation, we estimated the endogenous amounts of both Gip1 and Gβγ proteins. Since Gβ alone is unstable in Gγ-null cells (Fig. 3c), the amount of Gβ could be equivalent to the amount of Gβγ. Our calculations yielded 240,000 molecules of Gβγ in total and 60,000 molecules in the cytosol per cell (Fig. 4b, Supplementary Figure 6d). On the other hand, there are 157,000 molecules of Gip1 per cell, predominantly in the cytosol (Fig. 4b, Supplementary Figure 6d). Therefore, the cytosol of a cell contains slightly more Gip1 than Gβγ, suggesting that most cytosolic G proteins form complexes with Gip1.

*Dictyostelium* cells contain predictable prenylated proteins other than Gβγ. For example, there are 15 Ras homologues in which RasB, C, D, G, S are likely to have geranylgeranyl modifications according to the primary sequences of the CAAX motif[33]. However, Gip1 was not able to pull down Ras proteins under conditions of interaction with Gβγ (Supplementary Figure 6e). We also studied the interactions of proteins with CAAL motifs (which can undergo geranylgeranyl modification), such as RasG, Rac1A, and Rap1, with Gip1 (Supplementary Figure 6f). Gγ coprecipitated with Gip1, but RasG, Rac1A, and Rap1 did not

(Supplementary Figure 6g). These results show that Gip1 binds preferentially to G protein.

**Gip1 cavity is essential for G protein interactions**. To further investigate the importance of the hydrophobic cavity for complex formation, we analysed the structure–function relationship of Gip1 by introducing steric hindrance into the cavity. On the basis of structural information, the CASTp server selected 40 residues exposed to the interior surface of the cavity[34] (Supplementary Figure 7a). Among them, 24 leucine, isoleucine, and valine residues were replaced with tryptophan. The effect on the binding ability of G proteins was assessed by observing subcellular Gα2 localization using tetramethyl rhodamine (TMR) labelling (Supplementary Figure 7b). We found that 19 of the 24 mutants impaired cytosolic Gα2 localization to varying degrees (Fig. 5a, b, Supplementary Figure 7c). Those residues were distributed throughout the cavity, and nine of them appeared to have hydrophobic interactions with the bound phospholipid (Fig. 5a, Supplementary Figure 7a). We noticed that the cells lacking Gβ had lower amounts of cellular Gα2 proteins, which are present mostly in the cytosolic enriched fraction (Supplementary Figure 7d). This result suggests that Gα2 localization and stability depend heavily on normal Gβγ. As expected from the Gα2 localization results, Gβγ localization, as visualized by TMR-Gγ, was impaired in *gip1Δ* cells expressing Gip1 with Trp mutations at positions 306, 166, 300, 190 and 211 (Fig. 5b, Supplementary Figure 7e). These defects were confirmed by a biochemical pull-

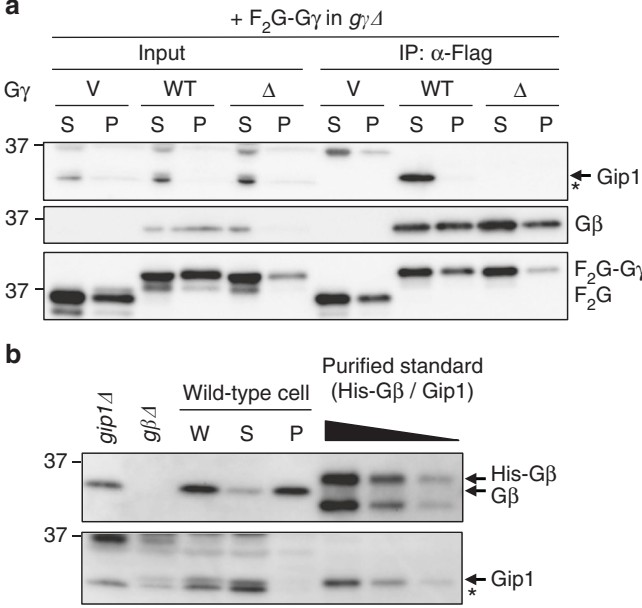

**Fig. 4** Subcellular localization of endogenous Gip1 and Gβ. **a** Preferential complex formation between G proteins and Gip1. Cells were fractionated, and each fraction was used for the IP of $F_2G$ or $F_2G$-tagged Gγ. The indicated proteins were visualized by immunoblotting using anti-Gip1, Gβ, and Flag antibodies. **b** The amount of endogenous Gβ and Gip1. Cells were fractionated to obtain whole-cell extract (W), supernatant (S), and precipitant (P), and their containing proteins were estimated in comparison to purified protein standards. The bands indicated by arrows represent His-Gβ, Gβ, and Gip1. Bands with an asterisk denote nonspecific bands

down assay with Gip1. Gip1 mutants with impaired G protein localization were not able to co-precipitate G proteins, while Gip1 (wild-type) could (Fig. 5c). These results strongly suggest that the hydrophobic cavity can be occupied by the geranylgeranyl moiety of Gγ to sequester G proteins in the cytosol.

**Function of hydrogen bonds at the cavity entrance.** We further mutated each residue of Gip1(146–310) to alanine and mapped the effect on cytosolic Gα2 localization in the crystal structure (Fig. 6a, Supplementary Figure 8a, b). The structure shows that mutations that cause defects in the cytosolic sequestration of Gα2 are mainly located on the α3 and α6 helices. Asp208 was one of the residues affected most severely by the substitutions and formed hydrogen bonds with both the side chain of Arg212 and the main chain nitrogen of Leu308 at the rim of the cavity entrance, which connected the C-terminal tail with the α3 helix (Fig. 2d). Moreover, Glu307 in the C terminus was also influenced by alanine substitution, and its side chain was directed towards the solvent (Fig. 2d, Supplementary Figure 3d). The importance of the C terminus was confirmed by deleting a.a. 304–310 (Gip1(ΔC-tail)). Similar to Gip1(D208A), Gip1(ΔC-tail) exhibited impaired interaction with G proteins (Fig. 6b, c). We next examined how the C terminus is implicated in the interaction by the in vitro-binding assay as in Fig. 3d. In contrast to the results in cells, which showed a binding defect, Gip1(ΔC-tail) bound to Gβγ as well as wild-type Gip1 did (Fig. 6d, Supplementary Figure 8c). Furthermore, when the CAAX box from *Dictyostelium* RasG was fused with GFP, these proteins bound to both wild-type and ΔC-tail Gip1 (Fig. 6d, Supplementary Figure 8c). That is, the loss of function of Gip1(ΔC-tail) is independent of the geranylgeranyl moiety. Taken together, these results

suggest that the characteristic location and configuration of the Gip1 C-terminal tail is necessary for Gip1 function.

**Significance of complex formation in chemotaxis.** To determine whether the complex formation of G proteins with Gip1 through the hydrophobic cavity plays a role in chemotactic signalling, we examined the chemotaxis ability of a series of mutant cells that exhibited defects in the cytosolic sequestration of G proteins. When 100 μM of cAMP was applied from a micropipette, *gip1Δ* cells rescued by Gip1 (wild-type) reached the source of the cAMP gradients, while cells carrying the vector alone had reduced chemotactic ability at the region near the tip of the micropipette. Consistent with a previous report, these phenotypes showed that Gip1 is required for chemotaxis at high concentrations[20] (Fig. 7a, b, Supplementary Movies 3 and 4). The cytosolic sequestration mutants Gip1(D208A) and Gip1(ΔC-tail) exhibited chemotaxis defects similar to those of *gip1Δ* cells (Fig. 7a, b, Supplementary Movies 5 and 6). These defects were also confirmed by a small population assay at different cAMP concentrations. While all *gip1Δ* cell lines harbouring only vector, Gip1(wild-type), Gip1(D208A), or Gip1(ΔC-tail) exhibited similar responses to droplets containing the lower cAMP concentrations, only Gip1(wild-type) expression maintained the chemotactic ability of *gip1Δ* cells at cAMP concentrations higher than 3 μM (Fig. 7c). In addition to these mutants, we examined the chemotactic ability of Trp-substituted mutants inside the hydrophobic cavity of Gip1 (Fig. 5a, b). Neither Gip1 (I306W, I166W, L300W, V190W and L211W) nor the vector was able to rescue the chemotactic defects of *gip1Δ* cells at the higher cAMP concentrations (Supplementary Figure 9). Moreover, the Gip1 mutants were not able to induce the membrane translocation of Gα2 upon cAMP stimulation (Fig. 7d, e). Because the Gip1 mutants analysed were normally expressed without obvious degradation (Fig. 5b, c and 6b, c), neither stability nor residues other than the mutated residues contributed to the defects in wide-range chemotaxis. These results demonstrate that Gip1 plays critical roles in wide-range chemotaxis through the sequestration and translocation of G proteins.

## Discussion

Lipid-modified proteins, represented by small G proteins, are spatially regulated by their lipid-bound proteins, known as solubilization factors[35], but no such regulatory mechanism is known for heterotrimeric G proteins. In this report, we determined the crystal structure of the C-terminal region of Gip1 and suggested the significance of the hydrophobic cavity for its interaction with G proteins and effective chemotaxis. The G protein-binding region of Gip1 contains a central hydrophobic cavity. Steric blockade by tryptophan substitution of the inner cavity abrogates Gip1 function. Geranylgeranyl modification of the Gγ subunit is essential for complex formation with Gip1. Together, these data suggest that Gip1 accommodates the geranylgeranyl modification of Gγ inside the hydrophobic cavity. In previous work, we observed the presence of a cytosolic fraction of G proteins but could not resolve how hydrophobic lipid modifications are stabilized in the cytosol[20]. In the current study, we found that removal of the prenyl modification of Gγ, resulting in Gγ(ΔCAAX), altered G protein membrane localization to the cytosol. This finding suggests that the hydrophobicity of G proteins is countered by binding to the hydrophobic cavity of the surface-charged Gip1, thereby stabilizing cytosolic localization.

Gip1 serves as a solubilization factor for the spatial regulation of heterotrimeric G proteins. Solubilization factors facilitate the cytosolic solubility of lipid-modified proteins (cargo) for appropriate localization to intracellular compartments. Its mechanism requires a common structural feature, that is, a hydrophobic

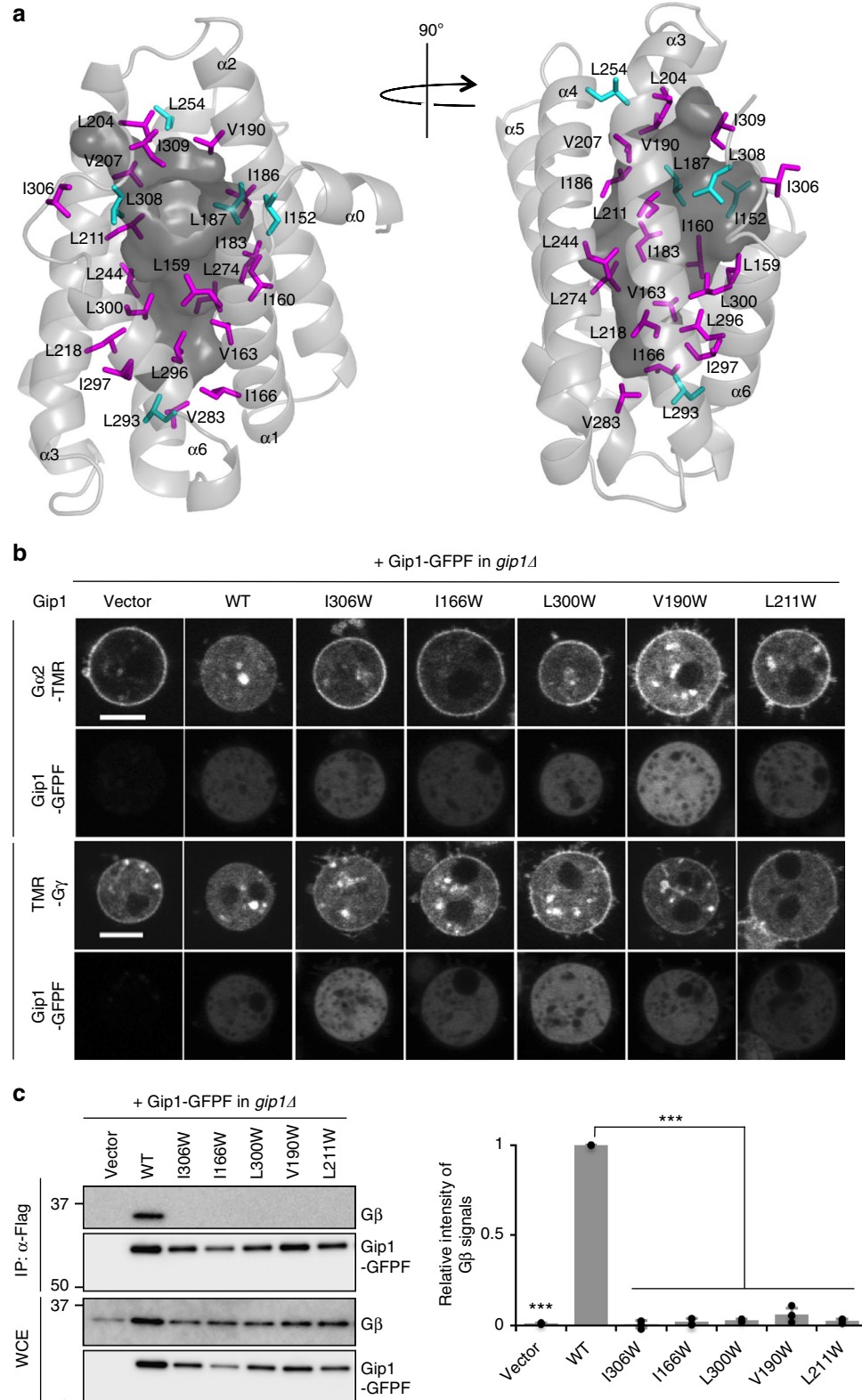

**Fig. 5** Tryptophan mutagenesis to induce steric blockade in the cavity. **a** Representation of tryptophan-mutated residues. Mutated residues are shown as stick models. Magenta and cyan colours show the residues that lead to cytosolic Gα reduced and the residues with no effects, respectively. The surface of the cavity is coloured dark grey. **b** Subcellular localization of Gα2 and Gγ labelled with TMR and Gip1-GFPF in a living cell in the presence of LatA. Representative images of the top 25% of severely impaired mutants are shown. Scale bar, 5 μm. **c** Co-immunoprecipitation of Gip1 with Gβγ. The same mutants shown in **b** were analysed. The data were normalized relative to the band intensities of wild-type Gip1 and are presented as the mean ± SD of at least three independent experiments (*n* = 5 (Vector, WT), 4 (I306W, I166W, V190W), and 3 (L300W, L211W), ***P < 0.001 versus wild-type, two-tailed unpaired Student's *t*-test)

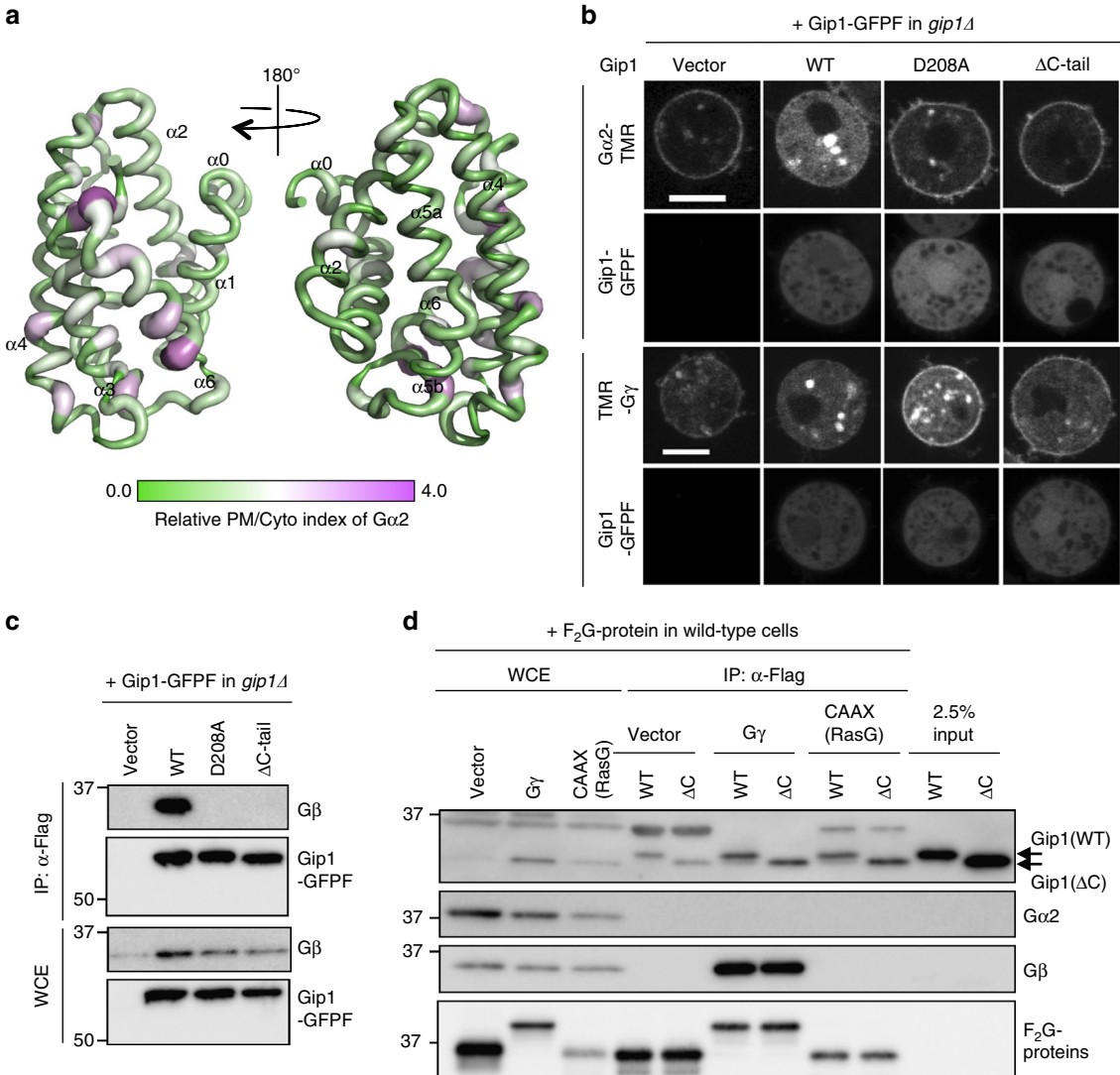

**Fig. 6** Comprehensive alanine mutagenesis scan of Gip1(146–310). **a** Structural mapping of the alanine mutagenesis scan of Gip1(146–310). Relative plasma membrane (PM)/cytosol (Cyto) indexes were normalized by the wild-type value and mapped onto the structure. The values range from purple (strong) to green (weak), represented by both the colour and the thickness of the ribbon diagram. **b** Subcellular localization of Gα2 and Gγ labelled with TMR and Gip1-GFPF in the presence of LatA. ΔC-tail, deletion of a.a. 304–310 of Gip1. Scale bar, 5 μm. **c** Co-immunoprecipitation of Gip1 with Gβγ. **d** In vitro interactions of purified Gip1. Flag-tagged GFP (F₂G) as a vector (V), F₂G-Gβγ, and F₂G containing the CAAX box (a.a. 178–189) from RasG were bound to beads and incubated with purified full-length Gip1 (WT) or C terminus-deleted Gip1, a.a. 1–303 (ΔC)

cavity or pocket to incorporate the lipid modification of the cargo, such as isoprenylation or myristoylation. Solubilization factors, including guanine nucleotide dissociation inhibitor[36] (GDI), are summarized in Supplementary Table 2. RhoGDI, phosphodiesterase-δ (PDEδ) and UNC119 exhibit a hydrophobic cavity composed by β-sheets, while RabGDI has a hydrophobic pocket on its structure. In contrast to these proteins, Gip1 exhibits a hydrophobic cavity constructed of six α-helices. Also, its cavity has a relatively large volume compared to that of RhoGDI, PDEδ and UNC119. Gip1 exhibits better binding to the heterotrimeric form of G proteins than to the βγ subunit alone[20]. The size of the Gip1 cavity might be adapted to accommodate the lipid modifications of both the α and βγ subunits.

Solubilization factors have been proposed to bind cargo proteins in two distinct ways. The ligand-free apo forms of RhoGDI and PDEδ showed closed and open cavity conformations, respectively[37]. The interaction of closed RhoGDI with Rho family proteins at the plasma membrane induces the opening of the cavity to bind the lipid modification of Rho. In contrast, cytosolic

PDEδ captures its cargo proteins, photoreceptor phosphodiesterase and KRas4B, upon spontaneous dissociation from the plasma membrane[37,38]. Similar to PDEδ, Gip1 is likely to sequester spontaneously dissociated G proteins. First, Gip1 occurs and forms complexes with G protein primarily in the cytosol (Fig. 4a, b). Second, our solved Gip1 structure contained bacterial glycerophospholipids (Fig. 1a, b), which suggests that the hydrophobic cavity is stably formed even without G proteins as cargo. However, interestingly, the phenotypes resulting from the depletion of PDEδ and of Gip1 are different. PDEδ knockdown eliminated KRas4B localization at the plasma membrane[38,39], while Gip1 deletion depleted G proteins in the cytosol and accumulated them at the plasma membrane[20] (Figs. 5b and 6b). The difference suggests a unique mechanism consisting of a Gip1-mediated spatial cycle of G protein. Additional structural analysis of Gip1 should provide evidence to illuminate the complete solubilization cycle.

The present study identified Gip1 as another type of prenyl-binding protein. The overall structure of Gip1 is similar to that of

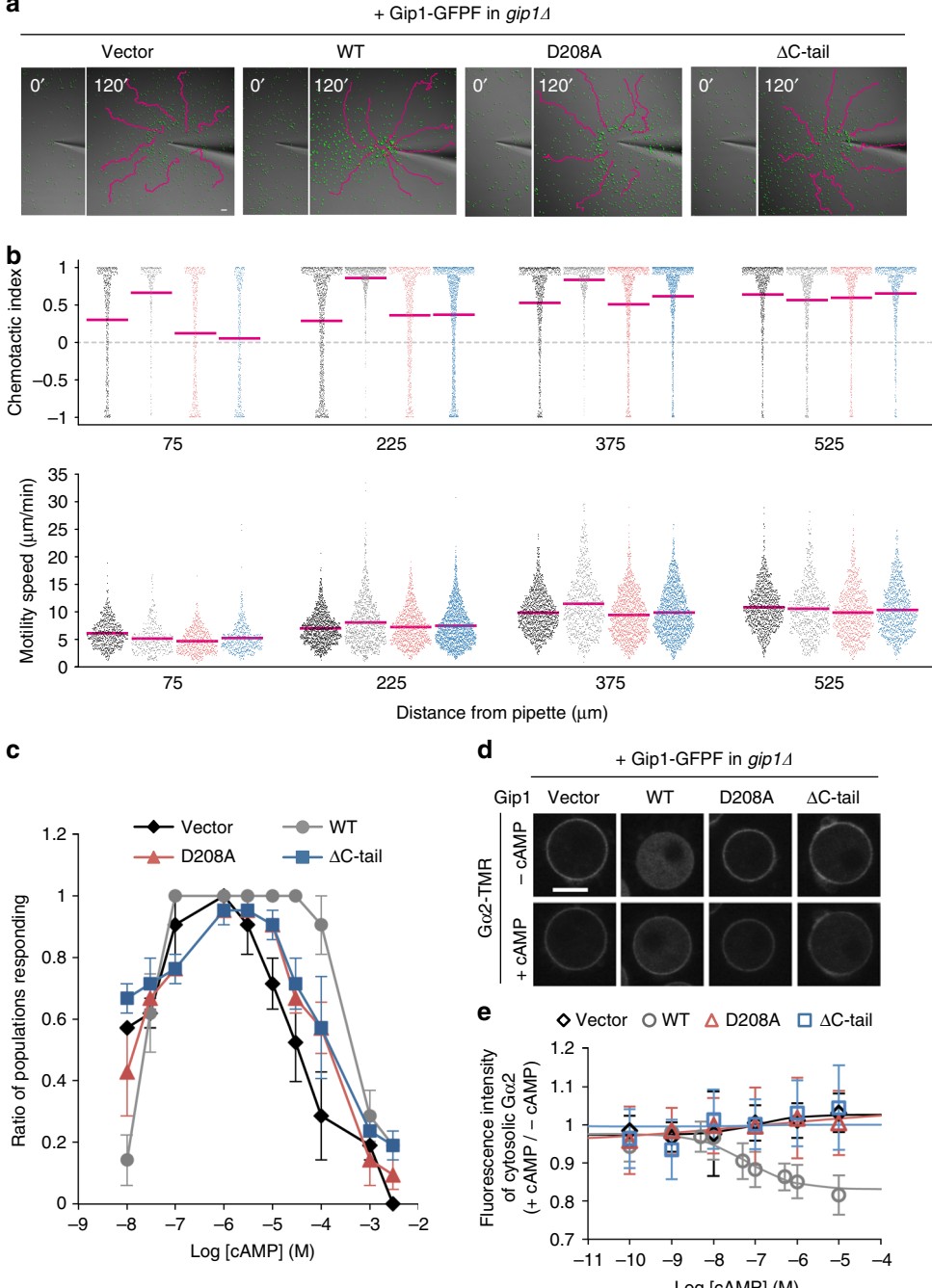

**Fig. 7** Contribution of Gip1–G protein complex formation to wide-range chemotaxis. **a** Chemotaxis of *gip1Δ* cells expressing Gip1 mutants. Cells (green) moved towards the tip of a micropipette filled with 100 μM cAMP. Representative images with cell trajectories (magenta lines) are shown before (0′) and 120 min after (120′) the start of the assay (see Supplementary Movies 3–6). Scale bar, 50 μm. **b** Chemotactic index (top) and motility speed (bottom) calculated from the assay in **a**. The magenta lines represent the mean (*n* = 285–1737 data points from at least 103 cells). Vector, WT, D208A, and ΔC-tail are shown in black, grey, red, and blue, respectively. **c** Chemotactic response to various cAMP concentrations. The data represent the mean ± SEM of three independent experiments. Vector, WT, D208A, and ΔC-tail are shown by the black-filled diamond, grey-filled circle, red-filled triangle, and blue-filled square, respectively. **d** Gα2 translocation upon cAMP stimulation in the presence of LatA. Images are from before (−cAMP) and after (+cAMP) 10 μM cAMP application. Scale bar, 5 μm. **e** Dose dependency of Gα2 translocation in response to different cAMP concentrations. The data represent the mean ± SD (*n* ≥ 60 cells). Vector, WT, D208A, and ΔC-tail are shown by the black-open diamond, grey-open circle, red-open triangle, and blue-open square, respectively

the TNFAIP8 family proteins, although the similarity in the primary sequences is low. TNFAIP8 proteins are assumed to bind lipid molecules, including phosphatidylinositol (3,4,5)-trisphosphate (PIP3), to perform functions related to immunity and tumourigenesis[26]. Our data suggest that some TNFAIP8 family proteins might target lipid modifications, including a prenyl moiety. TNFAIP8 is reported to bind Rac1 by an unknown mechanism[40]. Future work should consider the effect of the prenyl modification of Rac1 on its interaction with TNFAIP8. In spite of the presence of multiple prenyl-binding proteins in

eukaryotic cells, Gip1 specifically recognizes heterotrimeric G proteins, as shown in Supplementary Figure 6e–g. One reason for this specificity could be that Gip1 has an additional interaction site for heterotrimeric G proteins. Consistent with this idea, we identified Glu307 as an essential amino acid for G protein interaction. Since this side chain is directed towards the solvent (Fig. 2d), Glu307 may be involved in the inter-protein interaction with G proteins. Moreover, Ala substitution at Asp208 impaired Gip1 function (Fig. 6a–c). By hydrogen bonds to the α6 helix, Asp208 could retain the proper conformation of the C terminus (Fig. 2a–c). In fact, Gip1(ΔC-tail) is still able to bind to the geranylgeranyl moiety in vitro (Fig. 6d). These findings suggest that the complex formation involves not only the geranylgeranyl moiety on Gβγ but also contact via the C terminus of Gip1 in vivo. This theory is consistent with the idea that the periphery of the cavity entrance specifies the interacting molecules for the TNFAIP8 family[41].

Upon chemoattractant stimulation, G proteins are likely to be released from Gip1[20]. PDEδ, a solubilization factor, transports KRas4B from the plasma membrane to the endomembrane[42]. When the PDEδ-KRas4B complex encounters Arl2/3 at the endomembrane, PDEδ releases KRas4B to the target membrane by reducing the volume of the cavity[43]. This evidence suggests a mechanism in which the structural change in Gip1 could be required for Gip1-mediated G protein translocation. We obtained two different structures of Gip1 in terms of cavity shape (Fig. 2a–c). These properties suggest that the strength of the binding to G proteins might depend on the configuration of the cavity, as adjusted through a mobile α6 helix. Since the PH domain of Gip1 is required for the chemoattractant-triggered membrane translocation of cytosolic G proteins, the location of the PH domain in the Gip1 structure might be responsible for modulating the cavity configuration under chemotactic signalling[20] (Supplementary Figure 10). The PH domain could receive unknown factors independently of well-known chemotactic signalling, such as Ras, PIP3, and F-actin[20]. GPCR signalling is widely used in diverse biological phenomena. Moreover, mammalian TNFAIP8 has been reported to function in GPCR signalling[44]. Therefore, G protein shuttling-mediated spatiotemporal regulation might be conserved and shared as a common mechanism in other cell types[45–47]. The structural basis for the cytosolic sequestration of G proteins also provides mechanistic insights into the spatiotemporal regulation of lipid-modified proteins in living cells[48,49].

## Methods

**Cell growth and differentiation**. *Dictyostelium discoideum* AX2 was used as the parental strain. *gip1Δ*, *gγΔ* and *gβΔ* cells were described previously[20]. All cell lines used were derived from Masahiro Ueda's laboratory stocks. Cells were axenically grown in HL5 medium (Formedium) or on an SM plate (Formedium) with a *Klebsiella aerogenes* lawn at 22 °C. For preparing chemotactically competent cells, exponentially growing cells were collected and developed in developmental buffer (DB) consisting of phosphate buffer, 2 mM MgSO$_4$, and 0.2 mM CaCl$_2$ (pH 6.5) at $2 \times 10^7$ cells ml$^{-1}$ after 1 h of starvation, followed by the addition of 60 nM cAMP every 6 min for 4 h[51]. Cell phenotypes were identified on the *K. aerogenes* lawn or by plating on non-nutrient DB agar (1.5% agar in DB).

**Plasmids**. Plasmids were constructed by the In-Fusion technique (Clontech Laboratories) unless otherwise specified. pTX-Flag$_2$-GFP (pTX-F$_2$G), pTX-F$_2$G-Gγ, pJK1-Gip1-GFP-Flag, pJK1-Gα2 (at a.a. 90)-GFP-Flag, pHK12-(bla)-Gα2 (at a.a. 90)-Halo and pHK12-(bla)-Gγ-Halo were described previously[20]. pTX-F$_2$G-Gγ (ΔCAAX) was created by cloning the coding sequence of Gγ lacking the last four amino acids into pTX-F$_2$G. pTX-F$_2$G-CAAX(RasG) was created to express additional Cys-Thr-Leu-Leu, the CAAX box of RasG, after GFP. pTX-F$_2$G-RasG was created by cloning the coding sequence of RasG into pTX-F$_2$G. pTX-F$_2$G-Rac1A and pTX-F$_2$G-Rap1 were created by cloning the genomic region encoding each gene into pTX-F$_2$G. pJK1-Gip1(ΔC-tail)-GFP-Flag was created by cloning the coding sequence of Gip1(a.a. 1–303) into pJK1-GFP-Flag. pJK1-Gα2(G2A, C4G)-GFP-Flag was created by replacing the wild-type N terminus with mutated one

changing Gly to Ala at a.a. 2 and Cys to Gly at a.a. 4. All used primers are summarized in Supplementary Table 3.

**Immunoblotting**. Proteins were blotted onto a polyvinylidene difluoride membrane, which was then probed with the appropriate antibodies. Signals were visualized by chemiluminescence (Millipore), and images were acquired with ImageQuant LAS (GE Healthcare). A monoclonal anti-M2 antibody (Sigma-Aldrich, A8592, 1:25,000) was used to detect the Flag epitope. An anti-Ras antibody (#3965, 1:1,000) was purchased from Cell Signaling Technologies, USA. Rabbit polyclonal anti-Gβ (a.a. 35–51, 1:5,000) and -Gip1 (a.a. 96–110, 1:1000) antibodies were made in-house[20]. Rabbit polyclonal anti-Gα2 antibody (1:5000) was kindly provided by Dr. H. Kuwayama (Tsukuba University). Uncropped images are presented in Supplementary Figure 11.

**Overproduction and purification of Gip1(146–310)**. The plasmid vector pE-8HisSUMO-3C was created from pE-SUMOstar (LifeSensors) by elongating the poly-histidine chain from 6 to 8 units and inserting a PreScission cleavable site immediately after the SUMO sequence. The Gip1(146–310) fragment was amplified by PCR and cloned into pE-8HisSUMO-3C. The plasmid was transformed into Rosetta (DE3) competent *Escherichia coli* cells (New England Biolabs Japan). The cells were cultivated at 18 °C and harvested one day after isopropyl β-D-1-thiogalactopyranoside (IPTG) induction. The collected cells were resuspended in 20 mM HEPES-NaOH, pH 7.0, 350 mM NaCl, 30 mM imidazole, 5 mM MgCl$_2$, and 10 μg ml$^{-1}$ DNaseI and disrupted with a UD-201 ultrasonic disruptor (TOMY Seiko). Supernatant containing Gip1(146–310) was separated from the insoluble fraction by ultracentrifugation at 72,000 × *g*. Gip1(146–310) protein was purified by nickel-chelating resin (Ni-NTA, Qiagen), and the 8His-SUMO tag was cleaved by Turbo3C protease (Novagen) overnight at 4 °C. The protease reaction mixture was applied to the nickel-chelating resin. The flow-through fraction containing tag-free Gip1(146–310) was collected. Finally, Gip1(146–310) was purified by size-exclusion chromatography with Superdex 75 10/300 GL (GE Healthcare) in 20 mM HEPES-NaOH, pH 7.5, and 150 mM NaCl. Purified Gip1(146–310) was concentrated to 6.1 mg ml$^{-1}$ by ultrafiltration in Amicon Ultra with 3-kDa MWKO (Millipore), frozen with liquid nitrogen and stored at −80 °C.

**Crystallization and X-ray diffraction data collection**. We obtained crystals by the vapour-diffusion method. The crystallization reagent was composed of 14–20% (v/v) PEG 20,000 and 100 mM bicine, pH 8.0–9.0. Equal volumes of protein solution (Gip1(146–310) (6.1 mg ml$^{-1}$)) and reservoir solution were mixed and incubated at 20 °C. To reveal the structure of the complex between Gip1(146–310) and its ligand derivative, the crystals were also soaked in reagent containing 3 mM *N*-acetyl-*S*-geranylgeranyl-L-cysteine (Santa Cruz Biotechnology) and 0.025% DMSO, although the resulting structure did not contain the expected ligand (Form II). The crystals were soaked with reservoir solution containing 30% PEG 400 as a cryoprotectant and cryo-cooled with liquid nitrogen. Diffraction data were collected at BL26B1 and B2 of SPring-8 (Harima, Japan) at X-ray wavelengths of 1.0000 and 1.7000 Å for Form I and Form II, respectively. All X-ray experiments were performed under a cryostream at 100 K. Diffraction data were processed and scaled with HKL2000[51] (HKL Research) for Form I and XDS[52] for Form II.

**Structural determination and refinement**. The initial structure for the refinement of Gip1(146–310) was determined by molecular replacement (MR) using a poly-alanine model of a.a. 51–150 of TIPE2 protein (PDB 3F4M), which includes four of the six α-helices, as a search model. MR was performed with Phaser[53]. The side chains were modelled by using Autobuild[54] implemented in Phenix[55]. The obtained model structure was modified manually with Coot[56] and refined with Refmac5[57] implemented in the CCP4 program suite[58] and phenix.refine[59] implemented in Phenix. Ramachandran outliers were 0.61% for both Form I and Form II. To model the exogenous two ligands, PE and PG, the restraint files of dipalmitoyl-3-sn-phosphatidylethanolamine (PEF) and 1,2-dipalmitoyl-phosphatidylglycerol (LHG) from the CCP4 ligand library were used. Since the glycerophospholipid moiety occupied the same region, the head group was modelled by the alternate conformers of PEF and LHG with their common glycerophospholipid moiety. All crystallographic data and refinement statistics are summarized in Supplementary Table 1. All molecular graphics were produced with PyMOL (The PyMOL Molecular Graphics System, Version 1.8.3.2. Schrödinger, LLC.). The surface electron potential was calculated with the APBS tool[60].

**Structural alignment**. To compare the structural similarity, we conducted three-dimensional alignment by using the SUPERPOSE program implemented in the CCP4 program suite. The r.m.s. deviation of each Cα atom was calculated from the equivalent Cα atom of the template model structure[61].

**Determination of lipid extract from Gip1(146–310)**. To extract lipids, 0.5 ml of protein solution or buffer only was mixed with 1.9 ml of chloroform:methanol = 1:2 (v/v) mixture in a glass tube (chloroform:methanol:water = 1:2:0.8 (v/v/v))[62]. The mixture was vortexed and incubated at room temperature for 10 min. After incubation, we added 0.6 ml of chloroform and 0.6 ml of water and mixed before

centrifugation. We collected the bottom organic phase in another fresh glass tube and added 0.9 ml chloroform to the remaining aqueous solvent, followed by mixing and centrifugation. The bottom phase was collected and mixed with the first organic phase. After evaporation, the extracted lipids were dissolved in a small volume of chloroform and spotted 1 cm from the lower edge of an HPTLC Silica gel 60 plate (Millipore). The plate was placed in a chamber containing a developing solvent composed of chloroform:methanol:water = 65:25:4 (v/v/v) and incubated until the running front of the solvent was close to the upper edge. After incubation and air-drying, the plate was sprayed with $H_2SO_4$ and baked until the lipids became visible. As standards, L-α-phosphatidylethanolamine from egg yolk (Sigma-Aldrich) and L-α-phosphatidyl-DL-glycerol from egg yolk (Sigma-Aldrich) were developed in the same plate.

**Alanine scanning mutagenesis of Gip1.** Mutant Gip1 was created as follows. A designed primer including a codon for alanine substitution was synthesized as a forward primer, and its complementary sequence was synthesized as a reverse primer. The *gip1* gene was amplified as two fragments by PCR with the 5′-end primer of the *gip1* gene and the designed reverse primer and with the 3′-end primer of the gene and the designed forward primer. These two fragments were unified by fusion PCR with the 5′- and 3′-end primers of the *gip1* gene. This mutated gene fragment was cloned into a *Dictyostelium* expression vector and confirmed by DNA sequencing. The expression plasmid was introduced into *gip1Δ* cells containing Gα2-Halo.

For fluorescence imaging analysis of the Gα2-Halo distribution in vivo, starved cells in DB expressing the Gα2-Halo and Gip1 mutants were stained with TMR to visualize the Gα2-Halo and treated with 4 mM caffeine to prevent cAMP production for 30 min in light-shielding conditions at room temperature. Fluorescence imaging was performed with a confocal fluorescence microscope (A1, NIKON) after treatment with latrunculin A (LatA) for 10 min. The ratio of the fluorescence intensity between the cytosol and plasma membrane depended on the expression level of Gip1-GFPF, which was attributed to the interaction between Gα2 and Gip1. Based on these results, the binding activity of the Gip1 alanine mutants was roughly estimated by obtaining the ratio and relative plasma membrane (PM)/cytosol (Cyto) index ($A$). To obtain $A$, the ratio of the fluorescence intensity of Gα2 at the plasma membrane and in the cytosol was plotted against the Gip1-GFPF intensity, and the plots were fitted by a hyperbolic curve ($f(x) = A/x + C$). $C$ is a constant with a fixed value obtained from the results for wild-type Gip1-GFPF (Supplementary Figure 8a, b).

**Tryptophan mutagenesis scan of Gip1.** Surface-exposed residues in the cavity were identified by using the CASTp 3.0 server[34] with a spherical probe of 1.5 Å. Among the identified 40 residues, we selected 24 amino acids with hydrophobic side chains (leucine, isoleucine and valine) as candidates for tryptophan scanning. Mutants were constructed as described in the alanine scan mutagenesis of Gip1.

Cells were starved in DB for 2 h at 21 °C and stained with TMR ligands through Halo-tag for 30 min under light-shielded conditions at room temperature. Stained cells were observed with a confocal fluorescence microscope (FLUOVIEW FV1000, Olympus) after treatment with 5 μM LatA for 15 min. The fluorescence intensities of TMR at the plasma membrane and in the cytosol and of Gip1-GFP in the cytosol were calculated by ImageJ[63]. The ratio of the fluorescence intensity of TMR at the plasma membrane and in the cytosol was plotted against the Gip1-GFPF intensity. We selected cells that expressed Gip1-GFP and had a fluorescence intensity between 4000 and 6000 and calculated the mean and SD of the ratio of the fluorescence intensity of Gα2-Halo and Halo-Gγ (Supplementary Figure 7b).

**Pull-down assay.** Cells expressing Flag-tagged proteins were washed with DB and starved in phosphate magnesium (PM) buffer (5 mM Na/KPO$_4$ and 2 mM MgSO$_4$, pH 6.5) at a density of $2 \times 10^7$ cells ml$^{-1}$ for 2 h at 21 °C. The cells were lysed on ice in CHAPS buffer consisting of 40 mM HEPES (pH 7.5), 120 mM NaCl, 20 mM NaF, 2 mM Na$_3$VO$_4$, 20 mM sodium pyrophosphate, 0.3% CHAPS, and complete EDTA-free protease inhibitor (Roche) at a density of $4 \times 10^7$ cells ml$^{-1}$ followed by centrifugation to remove debris. The supernatants were incubated with anti-Flag M2 beads (Sigma-Aldrich, F2426) at 4 °C for 1.5 h followed by washing with CHAPS buffer. Protein samples were prepared by boiling the beads in 1× sodium dodecyl sulphate (SDS) sample buffer.

**Competitive assay.** The extracts of cells expressing GFP-Flag-tagged Gip1 were prepared as described in the pull-down assay. After incubation with the supernatants, anti-Flag M2 beads were washed with CHAPS buffer and incubated with CHAPS buffer at 4 °C for 1 h in the presence of 0, 10, 20, 40, and 100 μM of geranylgeranyl pyrophosphate (Sigma-Aldrich), 100 μM of farnesyl pyrophosphate (Sigma-Aldrich), or 100 μM of myristic acid (nacalai tesque) with 3.5% methanol and 1.5 μM NH$_3$. The beads were then washed with CHAPS buffer and boiled in 1× SDS sample buffer for the preparation of protein samples. The amount of Gip1-GFPF bound to M2 beads was quantified by immunoblotting with an anti-Flag antibody in comparison to carboxy-terminal DYKDDDK-BAP (BAP-Flag, Wako) as a standard.

**Fractionation assay.** For this assay, *gγΔ* cells expressing F$_2$G, F$_2$G-Gγ(WT), or F$_2$G-Gγ(ΔCAAX) were washed with DB and starved in PM buffer at a density of $2 \times 10^7$ cells ml$^{-1}$ for 2 h at 21 °C. Afterwards, the cells were resuspended at a density of $8 \times 10^7$ cells ml$^{-1}$, mixed with the same volume of basal buffer (20 mM Tris-HCl (pH 8.0), 2 mM MgSO$_4$), and fractionated by passing through Nuclepore Track-Etched Membranes (Whatman). The resulting cell lysate was centrifuged at $20,400 \times g$ for 1 min to separate the supernatant and precipitant. For the pull-down assay, the volumes of the supernatant and precipitant were equalized with 2× CHAPS buffer and 1× CHAPS buffer, respectively. Every sample was then centrifuged at $20,400 \times g$ for 15 min to remove insoluble debris, followed by the pull-down assay.

**In vitro-binding assay for Gγ activity.** 6His-SUMO-tagged Gip1(1–310) was overproduced in Rosetta (DE3) competent *E. coli* cells by cultivation overnight at 18 °C with 0.1 mM IPTG. The recombinant protein was purified by nickel-affinity chromatography with HisTrap HP (GE Healthcare), followed by size-exclusion chromatography with Superdex 200 Increase 10/300 GL (GE Healthcare). After the cleavage of 6His-SUMO-tag with SUMOstar protease (LifeSensors) overnight at 4 °C, the tag was removed by nickel-affinity chromatography. The flow-through fraction was applied to size-exclusion chromatography with Superdex 200 Increase 10/300 GL in 10 mM Tris-HCl, pH 8.0, 500 mM NaCl, and enriched samples at 10 mg ml$^{-1}$ were stored at −80 °C. *Dictyostelium* cells expressing vector or Flag-tagged Gγ(WT or ΔCAAX) were starved in PM buffer at a density of $2 \times 10^7$ cells ml$^{-1}$ for 2 h at 21 °C and lysed with NP40 buffer containing 40 mM HEPES (pH 7.5), 300 mM NaCl, 1% NP40 and complete EDTA-free protease inhibitor (Roche) at a density of $4 \times 10^7$ cells ml$^{-1}$, followed by centrifugation to remove debris. The supernatants were incubated with anti-Flag M2 beads at 4 °C for 1 h, washed with NP40 buffer and rinsed with CHAPS buffer. CHAPS buffer containing recombinant Gip1 and bovine serum albumin (BSA) was added to the rinsed beads and incubated for 30 min on ice followed by washing with CHAPS buffer. Protein samples were prepared by boiling the beads in 1× SDS sample buffer.

**In vitro-binding assay for Gip1 activity.** To compare the activity between Gip1 (1–310; WT) and Gip1(1–303; ΔC-tail), the corresponding regions of the *gip1* gene were amplified by PCR and cloned into the pE-8HisSUMO-3C plasmid. The plasmids were transformed into Rosetta (DE3) *E. coli* cells, which were cultivated at 18 °C and harvested one day after IPTG induction. The collected cells were resuspended in lysis buffer (20 mM HEPES (pH 7.0), 350 mM NaCl, 30 mM imidazole), disrupted by sonication, and followed by purification using nickel-chelating resin (Ni-NTA, Qiagen). After elution with lysis buffer containing 350 mM imidazole, the tag was cleaved with Turbo3C in 1 mM DTT overnight at 4 °C. Tag-free proteins were purified by collecting the flow-through from the nickel-chelating resin. The in vitro-binding assay was done using Gγ(WT) and the CAAX motif of RasG as the bead-binding proteins.

**Quantification of endogenous Gβ and Gip1.** His-tagged Gβ and Gip1(full-length) were used as standards for the quantification of endogenous Gβ and Gip1. His-tagged Gβ was overexpressed in Rosetta (DE3) cells. Cells were cultivated at 37 °C and harvested 2 h after 0.1 mM IPTG induction. Collected cells were disrupted by sonication in phosphate-buffered saline (PBS). After centrifugation at $20,400 \times g$, the precipitant was collected and boiled in 1× SDS sample buffer. As a control, bacterial cells before IPTG induction were subjected to the same His-tagged Gβ purification procedure. Recombinant Gip1 was prepared as a in vitro-binding assay for Gγ activity above. Protein concentrations were estimated in comparison with defined concentrations of BSA on a polyacrylamide gel stained with Coomassie Brilliant Blue.

The whole-cell extracts of *Dictyostelium* cells were fractionated into the supernatant and the precipitant. To estimate the amounts of endogenous proteins, purified Gβ and Gip1 with known concentrations were used as standards.

**Identification of lipid modifications by mass spectrometry.** Protein samples were separated by SDS polyacrylamide gel electrophoresis. Protein bands were visualized by Coomassie Brilliant Blue staining and excised, and the constituent proteins were digested with endoproteinase Glu-C for Gγ and with trypsin for Gα2. Peptides were separated by high-performance liquid chromatography and identified by tandem mass spectrometry.

**Small population chemotaxis assay.** Before the assay, cells were plated on a 35-mm plastic dish in DB buffer and incubated for 5 h to promote their development[20]. Approximately 3000 developed cells were placed on a 0.7% hydrophobic-treated agar plate (Wako, 010–08725; the agar powder was used after ten washes with Milli-Q water and dissolved in Milli-Q water containing 4 mM caffeine) in a 1-μl droplet of cell suspension in DB[50]. A drop of cAMP in DB was placed next to the cell droplet for 60 min. The distance between the centres of the two droplets was 2.5 mm. A droplet in which more than half of the total cells moved to the cAMP drop side was considered positive. The percentages of positive droplets were measured.

**Micropipette chemotaxis assay**. A total of $1 \times 10^5$ developed cells were seeded on a 27-mm glass-bottom dish (Iwaki). cAMP gradients were produced by a Femtotip microcapillary (Eppendorf) containing 100 μM cAMP and ATTO 633 (AD 633–21, ATTO-TEC) under a constant pressure of 10 hPa using a FemtoJet (Eppendorf). Images were acquired at 10-s intervals by confocal microscopy (A1, Nikon). Cells moving separately (i.e., cells not in contact with other cells) were tracked using G-Count (g-angstrom.com). Each trajectory was divided into short trajectories of 1-min intervals. The chemotaxis index and motility speed were analysed for each short trajectory. The chemotaxis index was the cosine of the angle formed by the intersection of the line connecting the starting and end points of movement with the line connecting the starting point and the micropipette. Motility speed was the total travelled distance divided by time. The analysed values of the trajectories were sorted by the distance from the end point of each trajectory to the tip of the micropipette and are shown as bar graphs.

**Statistical analyses**. All experiments were performed at least three times. To quantify the G protein localization, at least five cells were used. For the quantification of chemotaxis assay, at least 50 cells were analysed. Statistical analyses were carried out using Excel (Microsoft). The statistical significance was determined using a two-tailed unpaired Student's $t$-test. A $P$-value of less than 0.05 was considered statistically significant.

## Data availability

Data supporting the findings of this manuscript are available from the corresponding authors upon reasonable request. The atomic coordinates and structure factors have been deposited in the Worldwide Protein Data Bank under accession 5Z1N and 5Z39.

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

## Acknowledgements

We thank S. Taguchi and K. Tanabe for technical assistance. The *gip1Δ*, *gyΔ* and *gα2Δ* cell lines were provided by the National BioResource Project (NBRP)-nenkin. We also thank all members of the Ueda laboratory for discussions. The synchrotron radiation experiments were performed on beamlines BL26B1 and B2 at SPring-8 with the approval of the Japan Synchrotron Radiation Research Institute (JASRI) (Proposal Nos, 2015A1062, 2017A2503 and 2017B2503 to H.K. and 2017A2553 and 2017B2553 to K.T.) and of RIKEN (Proposal Nos, 20150100, 20160092 to H.K.). We thank the beamline staff for their support in the collection of X-ray data. This research was partially supported by the Japan Society for the Promotion of Science (JSPS) KAKENHI Grants 17K15105 (to Y.M.) and 17K07396 (to Y.K.) and by the Advanced Research and Development Programs for Medical Innovation (AMED-CREST) from the Japan Agency for Medical Research and Development, AMED JP17gm0910001 (to M.U.).

## Author contributions

T.M., H.K. and Y.K. conducted the experiments, designed the experiments and wrote the paper. Y.M. and K.T. conducted the experiments. A.N. wrote the paper. M.U. designed the experiments and wrote the paper.

## Additional information

**Competing interests:** The authors declare no competing interests.

