## [Peer Review File · Nature Communications]

Reviewers' Comments:

Reviewer #1:

Remarks to the Author:

In this MS, the authors reported the crystal structure of Gip1. Previously, the same group discovered that a novel regulator of G proteins, G-protein-interacting protein 1 (Gip1) and showed that Gip1 bind and sequester G protein subunits in cytosol. They found that activation of cAR1 also induces G-protein translocation from the cytosol the plasma membrane in a Gip1-dependent manner, suggesting that Gip1 regulates G-protein subunits translocating between plasma membrane and cytosol (Kamimura et al., 2016). The structure reported in this MS reveals the underlying molecular basis of Gip1's function. They showed that C-terminal Gip1 forms a hydrophobic cavity with six α -helices. Their results indicated that lipid modification on the G γ subunit is essential for Gbg interacting with Gip1. They previously indicated that Gip1 also interacts with G α subunits (Kamimura, 2016). However, it is not clear whether or how Gip1 interacts with G α 2 in this MS. They found that hydrogen bonds in the cavity of Gip1 are important for Gip1 function. They then showed that cells expressing Gip1 mutants that are defective in interacting with G-protein subunits, like *gip1*- cells, are defective in chemotaxis toward high dose cAMP.

It is an in-depth study that nicely reveals Gip1 functions at the molecular level. The experiments were well designed and results were clearly presented. I have some suggestions for them either to clarify the conclusions or to improve their manuscript.

Major points

1. Do G α 2 and Gbg form heterotrimers to interact with Gip1 in cytosol? Can G α s alone interact with Gip1? With the different lipid-modification and membrane localization of G α and Gbg, it is likely that G α and Gbg on the membrane translocate separately to the cytosol, where they interact with Gip1? This need to be clarified.
2. It is important to show that whether Gip1 interacts with G γ of the heterotrimeric G protein (G α bg) or dissociated Gbg equally or prefer Gbg? One way of examining above question is to do same pull-down assay as shown in Fig. 3c with/without high dose stimulation for a brief period of time.
3. For Fig. 4b, Fig. 5b, and Fig. 6d, the pair of proteins author should have shown is G γ -TMR and Gip1-GFP, although showing the pair of G α 2-TMR with Gip1-GFP is helpful to assume G γ /Gip1 membrane localization. The data shown in these figures rise another question that whether membrane localization of G α 2 subunit depends on Gbg.
4. In Fig. 3c and 3d, authors have shown that CAAX of G γ affects its interaction with Gip1. However, it is not clear whether CAAX of G γ affects the association of G α and Gbg subunits, which in turn affects the interaction between G protein and Gip1. Authors also did not distinguish which pool of G protein (on plasma membrane or in the cytosol) interacting with Gip1. A cleaner assay is to fractionate the cells then go through IP with cytosol sample or another way is to do IP assay with or without high dose cAMP stimulation.
5. Authors states that Gip1-dependent regulation of G-protein ensures wide-range gradient sensing in eukaryotic chemotaxis. However, other mechanisms of GPCR-mediated signaling events, such as receptor desensitization and the negative Ras regulator C2GAP1, are also essential for long-range chemotaxis. Gip1 regulates one of the signaling events controlling long-range chemotaxis. The authors should mention other studies and clarify the point.

Minor points:

1. What is the correlation between GG-pyroP (%) and the % of G γ geranylgeranylation
2. Graph labels of fig. 3d and fig. 4c are confusing. The authors need to provide a better explanation.
3. How can activation of cAR1 promote the translocation of G α Gbg cytosol to membrane? Can authors suggest some mechanisms?

Reviewer #2:

Remarks to the Author:

The paper by Miyawaga et al the authors present the crystal structure of Gip1 where they identify a hydrophobic pocket that they hypothesise to interact with the geranylgeranyl group of the Ggamma subunit and hence sequester the complex in the cytosol

In general this finding is important and the paper is well written however I have several suggestions and comments:

- 1- The authors should try to model the geranyl geranyl group in the hydrophobic pocket
- 2- The volume of the pocket should be reported and structural comparison to geranylgeranyl binding pockets such as in rabgdi and rho gdi should be provided
- 3- How much of Gip1 is in the cell can the authors comment on the stoichiometry and if there is enough of Gip1 to sequester the complex in the cytosol
- 4- I do not see experiments with endogenous Gip1 and Ggamma , are there no antibodies available for these proteins?
- 5- The specificity of the pocket towards different lipids should be studied. Did the authors try to compete the interaction of Gip1 with Ggamma with different lipids for example geranylgeranyl Vs farnesyl Vs myristoyl?
- 6- The specificity of Gip1 to G gamma Vs other geranylgeranylated proteins should be provided experimentally compare binding of Gip1 to other geranylgeranylated proteins such as Rabs and Rhos for example
- 7- In figure 5 the authors mutate Gip1 and look at the G alpha localisation as a readout of Gip1 interaction with Ggamma although it's a nice experiment however it is indirect and using the term "apparent dissociation constant " is over interpretation, I would suggest to use other description otherwise can be misleading
- 8- The authors identify D208 and the c term as important for localisation of g alpha, a detailed figure of how these regions are involved in the interaction with the geranylgeranyl is a must
- 9- Finally the authors show the importance of D208 in chemotaxis. However the authors show earlier that there are other residues that are also involved in the interaction of Gip1 with G gamma, can the authors show that these residues affect chemotaxis as well to support their model.

Minor comments :

- 1- The authors mention Ileu and Leu as amino acids with small side chains, that is not correct
- 2- Can the authors discuss their model compared to Rab and Rho GDIs for example in terms of extraction from membranes and solubilisation

Reviewer #3:

Remarks to the Author:

The manuscript "Structural basis of Gip1 for cytosolic sequestration of G-protein in wide-range chemotaxis" by Miyagawa et al. is a follow-up study of the recent report by this same group in PNAS. The PNAS paper identified Gip1 and suggested its function. This manuscript explores the structure-function relationships in Gip1 with a focus on how the Gip1 structure supports lipid binding. The manuscript begins with a structure determination of the C-terminal region of Gip1. This structure identifies that a hydrophobic pocket within this domain interacts with a lipid. The authors mutagenize the lipid-binding pocket and also remove the geranylgeranyl modification of Gg subunit by mutagenesis with the results suggesting that Gip1 binds to the geranylgeranyl of the Gg subunit. They then validated that this interaction is important for chemotaxis to cAMP using a knock out cell line that was complemented with empty vector, WT Gip1, or Gip1 variants that did not bind lipid in vitro. Together, the results are supportive of a mechanism where the interaction between Gip1 and the Gg subunit is via the geranylgeranyl modification and that this is important for chemotaxis. However, a primary concern is whether these results are sufficiently field-leading to warrant publication in a high impact journal. Or to put it another way, unlike the authors 2016 PNAS paper, which identified a new mechanism for regulating chemotaxis, the current manuscript is a closed-ended and self-contained study that does not seem to offer novel ideas that other

researchers could springboard off of into new areas of research.

Throughout, the manuscript is more difficult to read than an average manuscript, largely because of long and winding sentences with multiple take-home messages. Even the title was difficult for me to understand. The authors may want to consider recruiting the help of outside readers who can identify and help correct confusing syntax and who can also formalize the language and avoid slang. In many places, lack of precision in the language appears to decrease the accuracy of the narrative (in my opinion). There are numerous scientific proof-reading services available that can assist with this.

p. 6 second paragraph "Form I and Form II were almost identical, with an RMS deviation of 1.4 Å for the main chain". An RMS deviation of 1.4 Å is quite high for the same protein in two crystal forms, and I would strongly suggest removing the interpretation 'almost identical'. Although it appears that proteins in these two crystal forms have a global fold that is similar, the large rotations that the authors analyse next likely underlie this large conformational difference. This is perhaps the most interesting part of the structure to me and I would be enthusiastic about additional discussion from the authors on this conformational variability in the binding pocket and how it might contribute to mechanism.

p.7 – cavity mutations. Surface mutations of a protein are almost always stable, but mutations in a cavity or protein interior are less likely to be stable. Given that the Gip1 cavity seems to be unstable without something bound, it seems possible that the mutations introduced could impact protein stability or folding. Can the authors comment in the text on what they did to ensure that these were correctly folded?

Minor comments:

Throughout, "G protein" should not be hyphenated. "G protein-coupled receptor" is hyphenated correctly in the text.

Introduction, Paragraph 1, last sentence. "The structural basis of GPCR signaling has been extensively studied..." The authors may want to clarify that they mean the basis of GPCR signaling through G proteins. This is a very broad statement and the references cited to support this statement are unusual, consisting of one review and two papers from the Kobilka lab. Recognizing that one cannot reference all of the many papers in the field, the authors may want to focus on reviews and specifically indicate "as reviewed in (refs)" at the end of the sentence. Alternatively, if the authors have specific statements that are supported by the b2AR structures, the salient points should be explicitly indicated in the text.

The acronym for Gip1 is not defined in the main text. It is customary to define acronyms separately in both the abstract and the main text.

The abstract states that the structure of Gip1 is determined, but the text indicates that it is only the C-terminal region of Gip1 that was crystallized. The authors may want to consider correcting the abstract for accuracy.

p.6 second full paragraph starting with "Since Gip1 mainly binds to Gbg^{superscript} something". Whatever is supposed to be in the superscript is garbled.

The movies of the crystal structures show a duplication of some of the atoms of the lipid head group.

Reviewers' comments:

Reviewer #1 (Remarks to the Author):

In this MS, the authors reported the crystal structure of Gip1. Previously, the same group discovered that a novel regulator of G proteins, G-protein-interacting protein 1 (Gip1) and showed that Gip1 bind and sequester G protein subunits in cytosol. They found that activation of cAR1 also induces G-protein translocation from the cytosol the plasma membrane in a Gip1-dependent manner, suggesting that Gip1 regulates G-protein subunits translocating between plasma membrane and cytosol (Kamimura et al., 2016). The structure reported in this MS reveals the underlying molecular basis of Gip1's function. They showed that C-terminal Gip1 forms a hydrophobic cavity with six α -helices. Their results indicated that lipid modification on the G β subunit is essential for G $\beta\gamma$ interacting with Gip1. They previously indicated that Gip1 also interacts with G α subunits (Kamimura, 2016). However, it is not clear whether or how Gip1 interacts with G α_2 in this MS. They found that hydrogen bonds in the cavity of Gip1 are important for Gip1 function. They then showed that cells expressing Gip1 mutants that are defective in interacting with G-protein subunits, like gip1-cells, are defective in chemotaxis toward high dose cAMP.

It is an in-depth study that nicely reveals Gip1 functions at the molecular level. The experiments were well designed and results were clearly presented. I have some suggestions for them either to clarify the conclusions or to improve their manuscript.

Major points

1. Do G α_2 and G $\beta\gamma$ form heterotrimers to interact with Gip1 in cytosol? Can G α s alone interact with Gip1? With the different lipid-modification and membrane localization of G α and G $\beta\gamma$, it is likely that G α and G $\beta\gamma$ on the membrane translocate separately to the cytosol, where they interact with Gip1? This need to be clarified.
2. It is important to show that whether Gip1 interacts with G $\beta\gamma$ of the heterotrimeric G protein (G $\alpha\beta\gamma$) or dissociated G $\beta\gamma$ equally or prefer G $\beta\gamma$? One way of examining above question is to do same pull-down assay as shown in Fig. 3c with/without high dose stimulation for a brief period of time.
3. For Fig. 4b, Fig. 5b, and Fig. 6d, the pair of proteins author should have shown is G $\beta\gamma$ -TMR and Gip1-GFPF, although showing the pair of G α_2 -TMR with Gip1-GFPF is helpful to

assume G γ /Gip1 membrane localization. The data shown in these figures rise another question that whether membrane localization of Ga2 subunit depends on Gbg.

4. In Fig. 3c and 3d, authors have shown that CAAX of Gg affects its interaction with Gip1. However, it is not clear whether CAAX of Gg affects the association of Ga and Gbg subunits, which in turn affects the interaction between G protein and Gip1. Authors also did not distinguish which pool of G protein (on plasma membrane or in the cytosol) interacting with Gip1. A cleaner assay is to fractionate the cells then go through IP with cytosol sample or another way is to do IP assay with or without high dose cAMP stimulation.

5. Authors states that Gip1-dependent regulation of G-protein ensures wide-range gradient sensing in eukaryotic chemotaxis. However, other mechanisms of GPCR-mediated signaling events, such as receptor desensitization and the negative Ras regulator C2GAP1, are also essential for long-range chemotaxis. Gip1 regulates one of the signaling events controlling long-range chemotaxis. The authors should mention other studies and clarify the point.

Minor points:

1. What is the correlation between GG-pyroP (%) and the % of Gg geranylgeranylation.
2. Graph labels of fig. 3d and fig. 4c are confusing. The authors need to provide a better explanation.
3. How can activation of cAR1 promote the translocation of GaGbg cytosol to membrane?
Can authors suggest some mechanisms?

Reviewer #2 (Remarks to the Author):

The paper by Miyawaga et al the authors present the crystal structure of Gip1 where they identify a hydrophobic pocket that they hypothesise to interact with the geranylgeranyl group of the Ggamma subunit and hence sequester the complex in the cytosol

In general this finding is important and the paper is well written however I have several suggestions and comments:

- 1- The authors should try to model the geranyl geranyl group in the hydrophobic pocket

- 2- The volume of the pocket should be reported and structural comparison to geranylgeranyl binding pockets such as in rabgdi and rho gdi should be provided

- 3- How much of Gip1 is in the cell can the authors comment on the stoichiometry and if there is enough of Gip1 to sequester the complex in the cytosol

- 4- I do not see experiments with endogenous Gip1 and Ggamma , are there no antibodies available for these proteins?

- 5- The specificity of the pocket towards different lipids should be studied. Did the authors try to compete the interaction of Gip1 with Ggamma with different lipids for example geranylgeranyl Vs farnesyl Vs myristoyl?

- 6- The specificity of Gip1 to G gamma Vs other geranylgeranylated proteins should be provided experimentally compare binding of Gip1 to other geranylgeranylated proteins such as Rabs and Rhos for example

- 7- In figure 5 the authors mutate Gip1 and look at the G alpha localisation as a readout of Gip1 interaction with Ggamma although it's a nice experiment however it is indirect and using the term "apparent dissociation constant " is over interpretation, I would suggest to use other description otherwise can be misleading

- 8- The authors identify D208 and the c term as important for localisation of g alpha, a detailed figure of how these regions are involved in the interaction with the geranylgeranyl is a must

9- Finally the authors show the importance of D208 in chemotaxis. However the authors show earlier that there are other residues that are also involved in the interaction of Gip1 with G gamma, can the authors show that these residues affect chemotaxis as well to support their model.

Minor comments :

1- The authors mention Ileu and Leu as amino acids with small side chains, that is not correct

2- Can the authors discuss their model compared to Rab and Rho GDIs for example in terms of extraction from membranes and solubilisation

Reviewer #3 (Remarks to the Author):

The manuscript “Structural basis of Gip1 for cytosolic sequestration of G-protein in wide-range chemotaxis” by Miyagawa et al. is a follow-up study of the recent report by this same group in PNAS. The PNAS paper identified Gip1 and suggested its function. This manuscript explores the structure-function relationships in Gip1 with a focus on how the Gip1 structure supports lipid binding. The manuscript begins with a structure determination of the C-terminal region of Gip1. This structure identifies that a hydrophobic pocket within this domain interacts with a lipid. The authors mutagenize the lipid-binding pocket and also remove the geranylgeranyl modification of Gg subunit by mutagenesis with the results suggesting that Gip1 binds to the geranylgeranyl of the Gg subunit. They then validated that this interaction is important for chemotaxis to cAMP using a knock out cell line that was complemented with empty vector, WT Gip1, or Gip1 variants that did not bind lipid in vitro.

Together, the results are supportive of a mechanism where the interaction between Gip1 and the Gg subunit is via the geranylgeranyl modification and that this is important for chemotaxis. However, a primary concern is whether these results are sufficiently field-leading to warrant publication in a high impact journal. Or to put it another way, unlike the authors 2016 PNAS paper, which identified a new mechanism for regulating chemotaxis, the current manuscript is a closed-ended and self-contained study that does not seem to offer novel ideas that other researchers could springboard off of into new areas of research.

Throughout, the manuscript is more difficult to read than an average manuscript, largely because of long and winding sentences with multiple take-home messages. Even the title was difficult for me to understand. The authors may want to consider recruiting the help of outside readers who can identify and help correct confusing syntax and who can also formalize the language and avoid slang. In many places, lack of precision in the language appears to decrease the accuracy of the narrative (in my opinion). There are numerous scientific proof-reading services available that can assist with this.

p. 6 second paragraph “Form I and Form II were almost identical, with an RMS deviation of 1.4 Å for the main chain”. An RMS deviation of 1.4 Å is quite high for the same protein in two crystal forms, and I would strongly suggest removing the interpretation ‘almost identical’. Although it appears that proteins in these two crystal forms have a global fold that is similar, the large rotations that the authors analyse next likely underlie this large conformational

difference. This is perhaps the most interesting part of the structure to me and I would be enthusiastic about additional discussion from the authors on this conformational variability in the binding pocket and how it might contribute to mechanism.

p.7 – cavity mutations. Surface mutations of a protein are almost always stable, but mutations in a cavity or protein interior are less likely to be stable. Given that the G β 1 cavity seems to be unstable without something bound, it seems possible that the mutations introduced could impact protein stability or folding. Can the authors comment in the text on what they did to ensure that these were correctly folded?

Minor comments:

Throughout, “G protein” should not be hyphenated. “G protein-coupled receptor” is hyphenated correctly in the text.

Introduction, Paragraph 1, last sentence. “The structural basis of GPCR signaling has been extensively studied...” The authors may want to clarify that they mean the basis of GPCR signaling through G proteins. This is a very broad statement and the references cited to support this statement are unusual, consisting of one review and two papers from the Kobilka lab. Recognizing that one cannot reference all of the many papers in the field, the authors may want to focus on reviews and specifically indicate “as reviewed in (refs)” at the end of the sentence. Alternatively, if the authors have specific statements that are supported by the b2AR structures, the salient points should be explicitly indicated in the text.

The acronym for G β 1 is not defined in the main text. It is customary to define acronyms separately in both the abstract and the main text.

The abstract states that the structure of G β 1 is determined, but the text indicates that it is only the C-terminal region of G β 1 that was crystallized. The authors may want to consider correcting the abstract for accuracy.

p.6 second full paragraph starting with “Since G β 1 mainly binds to G β superscript something”. Whatever is supposed to be in the superscript is garbled.

The movies of the crystal structures show a duplication of some of the atoms of the lipid head group.

Responses to the editor's comments:

We would like to thank you and the reviewers for the supportive and constructive comments. In response to those comments, we provide new experiments and figures in the main text and new supplementary figures in the revised manuscript, as described below. In addition, following Reviewer #3's advice, we requested **Nature Research Editing Service** (URL <http://authorservices.springernature.com/>) to improve the usage of scientific language in our manuscript.

With these changes, we hope that our manuscript has become more convincing to readers.

In the submitted manuscript, our changes are marked by **green- text**.

List of changes

- **Figures**

Figure 1b (addition of connecting bonds)

Figure 3d (an *in vitro* binding assay between Gip1 and G protein)

Figure 3e (correction of the labelling of the lower bar graphs)

Figure 4a (cytosolic interaction between Gip1 and G protein)

Figure 4b (quantification of endogenous Gip1 and G protein)

Figure 5b (G γ localization in Gip1 Trp mutants)

Figure 5c (correction of the labelling of the right bar graphs)

Figure 6a (correction of the labelling of the colour bar)

Figure 6b (G γ localization in gip1 mutants)

Figure 6d (an *in vitro* binding assay between Gip1 and prenylated protein)

- **Supplementary Figures**

Supplementary Figure 3a (r.m.s. deviations between Form I and Form II)

Supplementary Figure 3b (comparison of a region with high r.m.s. deviations (a.a. 286-291) between Form I and Form II)

Supplementary Figure 3c (comparison of E307 in Form I and Form II)

Supplementary Figure 3e (comparison of cavity entrance in Form I and Form II)

Supplementary Figure 3g (comparison of a lipid in Form I and Form II)
Supplementary Figure 6b (quantification of Gip1 competitively targeted by geranylgeranyl-PP)
Supplementary Figure 6c (competitive assay with farnesyl-PP and myristic acid)
Supplementary Figure 6d (quantification of endogenous Gip1 and G protein)
Supplementary Figure 6e (no interaction between Gip1 and Ras)
Supplementary Figure 6f (localization of proteins with a CAAX motif)
Supplementary Figure 6g (no interaction between Gip1 and proteins with a CAAX motif)
Supplementary Figure 7d (G α 2 localization in g β - cells)
Supplementary Figure 7e (quantification of G γ localization in Gip1 Trp mutants)
Supplementary Figure 8c (quantification of Gip1 bound to prenylated protein *in vitro*)
Supplementary Figure 9 (small population assay of Gip1 Trp mutants)

- **Supplementary Movie**

Supplementary Movie 2 (addition of connecting bonds)

Our response to Reviewer #1:

We are grateful to Reviewer #1 for the critical comments and useful suggestions that have helped improve our manuscript. As indicated in the responses that follow, we have taken all the comments and suggestions into account in the revised version of our manuscript. We have put the comments made by the reviewer in italics with our responses below. Our changes in the revised manuscript are marked with **green- text**.

*In this MS, the authors reported the crystal structure of Gip1. Previously, the same group discovered that a novel regulator of G proteins, G-protein-interacting protein 1 (Gip1) and showed that Gip1 bind and sequester G protein subunits in cytosol. They found that activation of cAR1 also induces G-protein translocation from the cytosol the plasma membrane in a Gip1-dependent manner, suggesting that Gip1 regulates G-protein subunits translocating between plasma membrane and cytosol (Kamimura et al., 2016). The structure reported in this MS reveals the underlying molecular basis of Gip1's function. They showed that C-terminal Gip1 forms a hydrophobic cavity with six α -helices. Their results indicated that lipid modification on the G β subunit is essential for G β interacting with Gip1. They previously indicated that Gip1 also interacts with G α subunits (Kamimura, 2016). However, it is not clear whether or how Gip1 interacts with G α 2 in this MS. They found that hydrogen bonds in the cavity of Gip1 are important for Gip1 function. They then showed that cells expressing Gip1 mutants that are defective in interacting with G-protein subunits, like *gip1*- cells, are defective in chemotaxis toward high dose cAMP. It is an in-depth study that nicely reveals Gip1 functions at the molecular level. The experiments were well designed and results were clearly presented. I have some suggestions for them either to clarify the conclusions or to improve their manuscript.*

Our reply: We greatly appreciate this positive evaluation of our work. We have added detailed descriptions and several new experimental and analytical figures in the revised manuscript. We hope these additional descriptions are sufficient to resolve the concerns raised by the reviewer.

Major points

1. Do *Ga2* and *Gbg* form heterotrimers to interact with *Gip1* in cytosol? Can *Gas* alone interact with *Gip1*? With the different lipid-modification and membrane localization of *Ga* and *Gbg*, it is likely that *Ga* and *Gbg* on the membrane translocate separately to the cytosol, where they interact with *Gip1*? This need to be clarified.

Our reply: We thank the reviewer for raising this important question of how G proteins bind to *Gip1*. We investigated this issue in our previously published paper (Kamimura et al, PNAS, 2016). First, we pulled down *Gip1* from the extracts with no nucleotide, GDP or GTP γ S to change the G protein complex state. In this case, GDP addition maintained the interaction between *Gip1* and G proteins, whereas the addition of GTP γ S, a nonhydrolysable analogue of GTP, reduced the interaction (Fig. R1). Therefore, *Gip1* shows better binding to G proteins in a heterotrimeric state than either the α or $\beta\gamma$ subunit. Second, we also examined which subunit of G proteins binds to *Gip1* in a previous paper. Significant binding to the $G\beta\gamma$ subunit was observed (Fig. R2). Consistently, this current paper has shown that the geranylgeranyl modification on $G\gamma$ is essential for complex formation with *Gip1* (new Fig. 3). Furthermore, we revealed that *Gip1* was not able to bind to $G\alpha 2$ in cells lacking $G\beta\gamma$ in our previous PNAS paper (Fig. R3, red outline). Because most $G\alpha 2$ proteins are localized in the cytosol, although they are unstable in cells lacking $G\beta\gamma$ (new Supplementary Fig. 7d),

Fig. R1. *Gip1* preferentially binds to G proteins in a heterotrimeric form. Flag-tagged *Gip1* was pulled down in the presence of no nucleotide, 50 μ M GDP, or 50 μ M GTP γ S. The interaction was assessed by anti- $G\alpha 2$ and $G\beta$ immunoblots. (Reprinted from Fig. S1G in PNAS, **113**(16), 2016)

Fig. R2. *Gip1* preferentially binds to the $G\beta\gamma$ subunit. Each subunit of G proteins was purified on beads and mixed with bacterially purified *Gip1* to see the interaction. (Reprinted from a part of Fig. S1J in PNAS, **113**(16), 2016)

Gα2 alone cannot bind to Gip1. Finally, we confirmed that Gip1 is mostly localized in the cytosol (new Fig. 4, 5b, 6b). In response to Major point 4 by Reviewer #1, we performed experiments to test whether Gip1 interacts with G proteins in the cytosol and found that it does (new Fig. 4a). In summary, we conclude that Gip1 forms a complex with the G protein heterotrimer in the cytosol in the absence of cAMP stimulation. However, as the reviewer mentioned, cAMP stimulation causes Gα2 and Gβγ to dissociate from the membrane separately. Thus far, we do not know how the heterotrimer is rebuilt in the cytosol. The biochemical features described in the PNAS paper (Kamimura et al, PNAS, 2016) have been summarised in the introduction of the revised manuscript (page 6, lines 66-67).

2. It is important to show that whether Gip1 interacts with Gγ of the heterotrimeric G protein (Gαβγ) or dissociated Gβγ equally or prefer Gβγ? One way of examining above question is to do same pull-down assay as shown in Fig. 3c with/without high dose stimulation for a brief period of time.

Our reply: We thank the reviewer for highlighting this important issue regarding the interaction. As addressed in our response to Reviewer #1, Major point 1, Gip1 shows better binding to the heterotrimeric complex of G proteins than to the βγ subunit alone (Fig. R1).

3. For Fig. 4b, Fig. 5b, and Fig. 6d, the pair of proteins author should have shown is Gγ-TMR and Gip1-GFPF, although showing the pair of Gα2-TMR with Gip1-GFPF is helpful to assume Gγ/Gip1 membrane localization. The data shown in these figures rise another question that whether membrane localization of Gα2 subunit depends on Gβγ.

Fig. R3. Gip1 cannot bind to Gα2 subunit. Flag-tagged Gip1 was not able to co-precipitate with Gα2 in the absence of Gγ. The cell extracts were prepared from the indicated cells. The interaction was assessed by anti- Gα2 and Gβ immunoblots. (Reprinted from a part of Fig. S1K in PNAS, 113(16), 2016)

Our reply: We agree with this comment by the reviewer. Therefore, we observed G γ localization using Halo-tagged proteins in *gip1* mutants and obtained results consistent with those derived from G $\alpha 2$ localization. We have included these data in the new Fig. 5b, 6b, and Supplementary Fig. 7e of the revised manuscript (page 15, lines 217-220; page 16, lines 235-238).

The reviewer also asked whether membrane localization of G $\alpha 2$ depends on G $\beta\gamma$. Thus, we examined the localization of G $\alpha 2$ in cells lacking G β . Halo-tagged G $\alpha 2$ localized to the plasma membrane in wild-type cells but not in G β -null cells, although its expression level was low compared to that in wild-type cells. Furthermore, biochemical fractionation experiments confirmed that functional G $\beta\gamma$ subunits are required for the membrane localization as well as the stabilization of G $\alpha 2$ proteins. These data have been included in the new Supplementary Fig. 7d of the revised manuscript (page 15, lines 214-217).

4. In Fig. 3c and 3d, authors have shown that CAAX of G γ affects its interaction with Gip1. However, it is not clear whether CAAX of G γ affects the association of G α and G $\beta\gamma$ subunits, which in turn affects the interaction between G protein and Gip1. Authors also did not distinguish which pool of G protein (on plasma membrane or in the cytosol) interacting with Gip1. A cleaner assay is to fractionate the cells then go through IP with cytosol sample or another way is to do IP assay with or without high dose cAMP stimulation.

Our reply: We thank the reviewer for raising this important question. To show the direct involvement of the lipid modification of G γ in the interaction more clearly, we carried out an *in vitro* binding assay between Gip1 and G proteins. In brief, wild-type or Δ CAAX G $\beta\gamma$ proteins without G α subunits were purified on beads and mixed with Gip1 purified from bacteria. Gip1 binds to wild-type G $\beta\gamma$ but not to Δ CAAX G $\beta\gamma$ proteins. Therefore, the lipid modification of G γ through its CAAX motif ensures the interaction with Gip1 directly. These results have been included in the new Fig. 3d of the revised manuscript (page 12, lines 171-174).

The reviewer also asked which pool of G protein interacts with Gip1. Accordingly, we pulled down G γ proteins from the cytosolic or the membrane fraction. Gip1 was coprecipitated from the cytosolic pool of G γ proteins but not from the

membrane pool in wild-type cells. Furthermore, we also carried out the same experiment using $G\gamma(\Delta CAAX)$ proteins. These mutant proteins did not show co-immunoprecipitation with Gip1 even from the cytosol. These data show that the complex between Gip1 and G proteins is formed in the cytosol and depends on the lipid modification of the $G\gamma$ subunit (new Fig. 4a; page 13, lines 183-185).

5. Authors states that Gip1-dependent regulation of G-protein ensures wide-range gradient sensing in eukaryotic chemotaxis. However, other mechanisms of GPCR-mediated signaling events, such as receptor desensitization and the negative Ras regulator C2GAP1, are also essential for long-range chemotaxis. Gip1 regulates one of the signaling events controlling long-range chemotaxis. The authors should mention other studies and clarify the point.

Our reply: We appreciate the reviewer drawing our attention to other mechanisms for chemotactic wide-range sensing. In accordance with this comment, we have added other mechanisms along with Gip1-mediated regulation to the introduction of the revised manuscript (page 5, lines 55-60).

Minor points:

1. What is the correlation between GG-pyroP (%) and the % of Gg geranylgeranylation

Our reply: According to the reviewer's comment, we quantified geranylgeranyl pyrophosphate (GG-pyroP) and Gip1 proteins in the assay because GG-pyroP binds competitively to Gip1. In this case, 5% GG-pyroP (10 nmoles) completely inhibited 4 pmoles of Gip1 proteins. The data are included in the new Supplementary Fig. 6b (pages 12-13, lines 176-177).

2. Graph labels of fig. 3d and fig. 4c are confusing. The authors need to provide a better explanation.

Our reply: We appreciate the reviewer's careful reading of our manuscript. In accordance with this comment, we have changed the graph labels of the new Fig. 3e and

5c to “Relative intensity of G β signals”.

3. How can activation of cAR1 promote the translocation of GaGbg cytosol to membrane? Can authors suggest some mechanisms?

Our reply: We thank the reviewer for this interesting question. We showed that the translocation of G α 2G β γ is dependent on the PH domain of Gip1 and independent of PIP3, Ras, and F-actin in the previous PNAS paper. These results suggest that the translocation is likely regulated by an unknown chemotactic pathway. Since Gip1 is localized mainly in the cytosol, soluble substances could be produced upon cAMP stimulation and bind to its PH domains. However, we do not have any further evidence to speculate on what they are. Therefore, we have added only the evidence already shown to explain the model mechanism in the discussion of the revised manuscript (page 24, lines 358-359).

Our responses to Reviewer #2:

We are grateful to Reviewer #2 for the critical comments and useful suggestions that have helped improve our paper. We provide point-to-point answers to the comments below and have added detailed explanations to the revised manuscript. We present the comments made by the reviewer in italics with our responses below. Our changes in the manuscript are marked by **green- text**.

The paper by Miyawaga et al the authors present the crystal structure of Gip1 where they identify a hydrophobic pocket that they hypothesise to interact with the geranylgeranyl group of the Ggamma subunit and hence sequester the complex in the cytosol

In general this finding is important and the paper is well written however I have several suggestions and comments:

Our reply: We greatly appreciate this positive evaluation of our work. We have added detailed descriptions and several new experimental and analytical figures in the revised manuscript. We hope these additional descriptions are sufficient to resolve the concerns raised by the reviewer.

1- The authors should try to model the geranyl geranyl group in the hydrophobic pocket

Our reply: We thank the reviewer for this interesting comment. We ran the software Autodock Vina (Trott and Olson, J. Comp. Chem., 2010), leading to several models that incorporate the geranylgeranyl group in the hydrophobic cavity. Briefly, the model ligand was prepared by combining geranylgeranyl and C-terminal methylated cysteine to mimic geranylgeranylation at a CAAX motif of Gy. The model ligand was converted into an input file by the PRODRG server (Schüttelkopf and van Aalten, Acta Crystallogr, 2004) and placed manually near the Gip1 cavity entrance by using visual molecular dynamics (VMD) (Humphrey et al., J. Mol. Graph., 1996). Each structure resulted in a stability of -7 to -8 kcal mol⁻¹. We chose the structure in which the cysteine of the model ligand is located at the cavity entrance and exposed to solvent. As shown

in Fig. R4, the model ligand is not rigidly fixed in the cavity, probably due to the flexibility of its acyl chains, and there are several possible orientations in which the geranylgeranyl group can be incorporated into the cavity. Although the docking model clearly shows that the cavity size is sufficient to cover the geranylgeranyl moiety, we could not determine conclusively which form is physiologically relevant. Therefore, these data have not been included in the revised manuscript to avoid misleading the readers.

Fig. R4. The constructed models of Gip1 with a geranylgeranyl moiety. Form I and Form II of Gip1 are represented by ribbon diagrams in blue and orange, respectively. The surface of the cavity is shown in grey. The model ligand is shown in green.

2- The volume of the pocket should be reported and structural comparison to geranylgeranyl binding pockets such as in rabgdi and rho gdi should be provided

Our reply: We appreciate the reviewer's thoughtful comment. We calculated the pocket volumes of other solubilization factors, including RhoGDI, RabGDI, PDE δ , and UNC119, and compared these values and their structural features in the new Supplementary Table 2. The comparison shows the novelty of Gip1 as a solubilization factor and has been included in the discussion of the revised manuscript (pages 20-21, lines 294-307).

3- How much of Gip1 is in the cell can the authors comment on the stoichiometry and if there is enough of Gip1 to sequester the complex in the cytosol

Our reply: In accordance with the reviewer's suggestions, we quantified the endogenous Gip1 and G β proteins. The numbers of Gip1 and G β are approximately 240,000 and 150,000 molecules per cell, respectively. Because complex formation occurs in the cytosol (new Fig. 4a), we also calculated the cytosolic number of these proteins. Since approximately one-quarter of G β and almost all Gip1 are present in the cytosol, their cytosolic numbers are approximately 60,000 and 150,000 molecules, respectively. The amount of G β proteins is equivalent to the cellular amount of the G $\beta\gamma$ complex because G β alone is unstable in G γ -null *Dictyostelium* cells (new Fig. 3c). Thus, cytosolic Gip1 is present in excess of cytosolic G $\beta\gamma$. Taken together, this evidence suggests that most cytosolic G $\beta\gamma$ could form complexes with Gip1. These data have been included in the new Fig. 4b and the new Supplementary Fig. 6d of the revised manuscript (page 13, lines 185-192).

4- I do not see experiments with endogenous Gip1 and Ggamma, are there no antibodies available for these proteins?

Our reply: As stated above, we quantified the endogenous protein levels of Gip1 and G $\beta\gamma$ protein. We have included these data in the new Fig. 4b and the new Supplementary Fig. 6d of the revised manuscript (page 13, lines 185-192).

5- The specificity of the pocket towards different lipids should be studied. Did the authors try to compete the interaction of Gip1 with Ggamma with different lipids, for example geranylgeranyl Vs farnesyl Vs myristoyl?

Our reply: We thank the reviewer for this interesting question. To address this issue, we investigated the competitive effects of other lipids, including farnesyl pyrophosphate and myristic acid, on the complex between Gip1 and G proteins. Myristic acid was not able to interfere with the complex. Farnesyl pyrophosphate showed slight impairment of the Gip1-G protein complex although its effect was much less than that of

geranylgeranyl pyrophosphate. Therefore, the hydrophobic cavity has some specificity for the geranylgeranyl moiety. These results have been included in the new Supplementary Fig. 6c of the revised manuscript (page 13, lines 177-180).

6- *The specificity of Gip1 to G gamma Vs other geranylgeranylated proteins should be provided experimentally compare binding of Gip1 to other geranylgeranylated proteins such as Rabs and Rhos for example*

Our reply: Following the reviewer's suggestion, we investigated the binding of Gip1 to other possible geranylgeranylated proteins. First, we examined the interaction between Gip1 and Ras proteins in a pull-down assay. Ras proteins abundantly expressed in *Dictyostelium* cells, such as RasG, RasC, RasB, RasD, and RasS, have CAAX motifs for geranylgeranyl modification because X is Leu (Jiang et al, Chem. Rev., 2018). The pull-down fraction of Gip1 enriched G β but not Ras proteins (new Supplementary Fig. 6e). Next, we confirmed that Gip1 proteins were pulled down by G γ but not by RasG, Rac1A, and Rap1, all of which have CAAX motifs for geranylgeranyl modification. In this experiment, we omitted Rab and Rho homologues of *Dictyostelium* cells because many of them have CAAX motifs for farnesyl modification. According to these results, Gip1 shows specificity to G $\beta\gamma$ proteins. These data have been included in the new Supplementary Fig. 6f, g of the revised manuscript (page 14, lines 193-201).

7- *In figure 5 the authors mutate Gip1 and look at the G alpha localisation as a readout of Gip1 interaction with Ggamma although it's a nice experiment however it is indirect and using the term "apparent dissociation constant " is over interpretation, I would suggest to use other description otherwise can be misleading*

Our reply: We agree with the reviewer about this term. We have changed it to "cytosol to membrane index", as explained in the Methods section of the revised manuscript (page 33, lines 503-504).

8- *The authors identify D208 and the c term as important for localisation of g alpha, a detailed figure of how these regions are involved in the interaction with the geranylgeranyl is a must*

Our reply: We thank the reviewer for this interesting comment. To address this issue, we examined whether the C terminus of Gip1 interacts with the geranylgeranyl modification of G $\beta\gamma$ in an *in vitro* binding assay. Briefly, Gip1(WT) and Gip1(Δ C-tail) proteins were purified from bacteria. These proteins were mixed with Flagx2-GFP-tagged G $\beta\gamma$ (F2G-G $\beta\gamma$), and F2G-RasG(CAAX) where F2G is followed by the CAAX box from RasG. The experiment revealed that the binding ability of Gip1(Δ C-tail) to F2G-G $\beta\gamma$ and F2G-RasG(CAAX) is comparable to that Gip1(WT). These results suggest that Gip1(Δ C-tail) can bind to a geranylgeranyl modification. We noticed that Gip1(WT) binds to F2G-G $\beta\gamma$ and F2G-RasG(CAAX) almost equally. These results suggest that this experimental setting assesses only geranylgeranyl-dependent Gip1 binding ability. Therefore, as we originally discussed, the C terminus of Gip1 could be involved in inter-protein interactions with the G protein. These data have been included in the new Fig. 6d and the new Supplementary Fig. 8c of the revised manuscript (pages 16-17, lines 238-244; pages 22-23, lines 335-344).

9- Finally the authors show the importance of D208 in chemotaxis. However the authors show earlier that there are other residues that are also involved in the interaction of Gip1 with G gamma, can the authors show that these residues affect chemotaxis as well to support their model.

Our reply: In accordance with the reviewer's suggestions, we investigated the chemotactic ability of other mutants by a Trp mutation scan inside the hydrophobic cavity. As expected, Trp mutants were not able to complement the chemotactic defects of *gip1* Δ cells at 100 μ M cAMP. These data confirm the biological significance of the complex formation between Gip1 and G proteins and have been included in the new Supplementary Fig. 9 of the revised manuscript (page 18, lines 264-268).

Minor comments :

1- The authors mention Ileu and Leu as amino acids with small side chains, that is not correct

Our reply: In accordance with the reviewer's comment, we have deleted "with a small

side chain” from the sentence.

2- Can the authors discuss their model compared to Rab and Rho GDIs for example in terms of extraction from membranes and solubilisation

Our reply: Following the reviewer’s suggestions, we have added a discussion of the solubilization cycle of Gip1-mediated G proteins compared with PDE δ and RhoGDI with references to the discussion of the revised manuscript (pages 21-22, lines 308-324).

Our responses to Reviewer #3:

We are grateful to Reviewer #3 for the critical comments and useful suggestions that have helped improve our paper. As indicated in the responses that follow, we have addressed all the comments and suggestions in the revised version of our paper. We present the comments made by the reviewer in italics with our responses below. Our changes in the manuscript are marked by **green- text**.

The manuscript “Structural basis of Gip1 for cytosolic sequestration of G-protein in wide-range chemotaxis” by Miyagawa et al. is a follow-up study of the recent report by this same group in PNAS. The PNAS paper identified Gip1 and suggested its function. This manuscript explores the structure-function relationships in Gip1 with a focus on how the Gip1 structure supports lipid binding. The manuscript begins with a structure determination of the C-terminal region of Gip1. This structure identifies that a hydrophobic pocket within this domain interacts with a lipid. The authors mutagenize the lipid-binding pocket and also remove the geranylgeranyl modification of Gg subunit by mutagenesis with the results suggesting that Gip1 binds to the geranylgeranyl of the Gg subunit. They then validated that this interaction is important for chemotaxis to cAMP using a knock out cell line that was complemented with empty vector, WT Gip1, or Gip1 variants that did not bind lipid in vitro.

Together, the results are supportive of a mechanism where the interaction between Gip1 and the Gg subunit is via the geranylgeranyl modification and that this is important for chemotaxis. However, a primary concern is whether these results are sufficiently field-leading to warrant publication in a high impact journal. Or to put it another way, unlike the authors 2016 PNAS paper, which identified a new mechanism for regulating chemotaxis, the current manuscript is a closed-ended and self-contained study that does not seem to offer novel ideas that other researchers could springboard off of into new areas of research.

Our reply: We greatly appreciate this reviewer’s comment that our original manuscript did not express the novelty of this current work. As the reviewer agrees, we have shown that the interaction between Gip1 and the Gγ subunit occurs via geranylgeranyl

modification. We believe it is important to show that Gip1-mediated G protein shuttling is essential for wide-range chemotaxis. The structural analysis of Gip1 clearly elucidates its function as a solubilization factor for heterotrimeric G proteins and eliminates other possible enzymatic activities, such as GAP for G protein. Solubilization factors facilitate the inter-membrane distribution of lipid-modified proteins through the cytosol. Lipid-modified proteins include prenylated and myristoylated proteins, such as PDE, Ras, Rho, Rab, and Src, which influence a wide range of biological and medical phenomena. Therefore, the research field is expanding. Our study shows that heterotrimeric G proteins but not small G proteins are also subject to spatial regulation by a solubilization factor, Gip1. The mechanism of the intracellular spatial cycle is now being revealed by the structural-functional analysis of solubilization factors. Gip1 has both unique mechanisms and others shared with different proteins. Further Gip1 structural analysis will reveal the overall mechanism of G protein shuttling, providing insights into the general understanding of solubilization factors. Moreover, as we already stated in the original manuscript, this work will impact the molecular function of TNFAIP8 family proteins, which are deeply related to cancer, immunity, and other phenomena. We have added these statements in the revised manuscript make the importance of our work more convincing (pages 20-22, lines 294-324).

Throughout, the manuscript is more difficult to read than an average manuscript, largely because of long and winding sentences with multiple take-home messages. Even the title was difficult for me to understand. The authors may want to consider recruiting the help of outside readers who can identify and help correct confusing syntax and who can also formalize the language and avoid slang. In many places, lack of precision in the language appears to decrease the accuracy of the narrative (in my opinion). There are numerous scientific proof-reading services available that can assist with this.

Our reply: In accordance with the reviewer's comment, we asked Nature Research Editing Service for scientific proofreading of the revised manuscript.

p. 6 second paragraph "Form I and Form II were almost identical, with an RMS deviation of 1.4 Å for the main chain". An RMS deviation of 1.4 Å is quite high for

the same protein in two crystal forms, and I would strongly suggest removing the interpretation ‘almost identical’. Although it appears that proteins in these two crystal forms have a global fold that is similar, the large rotations that the authors analyse next likely underlie this large conformational difference. This is perhaps the most interesting part of the structure to me, and I would be enthusiastic about additional discussion from the authors on this conformational variability in the binding pocket and how it might contribute to mechanism.

Our reply: We thank the reviewer for this important comment and encouragement. To more precisely show the difference between Forms I and II, we calculated the r.m.s. deviation at each amino acid, as shown in the new Supplementary Fig. 3a. The analysis showed that $\alpha 1$ and $\alpha 6$ have high values compared with other helices, suggesting that these helices are flexible. Consistently, the $\alpha 6$ helix around a.a. 286-291 in Form I is kinked by penetration by water molecules, while $\alpha 6$ helix is tightly folded in Form II (new Supplementary Fig. 3b). In addition, as shown in the original manuscript, the large r.m.s. deviation differences around a.a. 152-159 ($\alpha 1$) and a.a. 298-303 ($\alpha 6$) are linked to rotational movements of the $\alpha 1$ and $\alpha 6$ helices and lead to a decrease in the cavity size of Form II (new Fig. 2a, b, c, Supplementary Fig. 3d), potentially in turn altering the location of a lipid inside the cavity (new Supplementary Fig. 3g). These are described in the revised manuscript (pages 9-10, lines 124-139).

To connect this conformational variability and the sequestering function of Gip1 to the solubilization of G protein in the cytosol, we compared and discussed the cavity size of solubilization factors in the new Supplementary Table 2 (pages 20-21, lines 294-307). Solubilization factors associate and dissociate with lipid-modified proteins depending on the volume of their hydrophobic cavities or pockets. This comparison leads to the hypothesis that the structural change in Gip1 is also required for Gip1-mediated G protein translocation. In addition, the location of the PH domain in the Gip1 structure might be responsible for modulating the cavity configuration through the $\alpha 1$ and/or $\alpha 6$ helices because the PH domain of Gip1 is essential for this translocation, as reported previously in our PNAS paper (Kamimura et al, PNAS, 2016) (pages 23-24, lines 346-359).

Moreover, we noticed that the C-terminal configuration of the $\alpha 6$ helix differs in Forms I and II. Our Ala scanning analysis showed that the C terminus is important

for G protein binding. First, E307A or the C-terminal deletion of Gip1 impaired G protein binding (new Fig. 6a, b, c; page 16, lines 227-238). Second, by forming hydrogen bonds to the $\alpha 6$ helix, D208 could maintain the proper composition of the C terminus (new Fig. 2a, b, c; page 10, lines 130-139). Interestingly, we have shown that Gip1(Δ C-tail) binds to the geranylgeranyl moiety *in vitro* even though there is no interaction with G proteins *in vivo* (new Fig. 6a-d; pages 16-17, lines 235-244). This behaviour suggests that the C terminus ensures complex formation through inter-protein interactions with G proteins. In particular, the side chain of E307 is directed towards the solvent in Form I but is oriented differently in Form II (new Supplementary Fig. 3c), which changes the surface charge at the top of Gip1. This evidence suggests that Form I is the conformation that binds with G protein. The rotational movements of the $\alpha 6$ helix also affect the C-terminal configuration of the binding form and result in the loss of E307-mediated interaction. Taken together, both the size of the cavity and the C-terminal configuration of Gip1 define the mechanism of the binding and release of G proteins. These points have been mentioned in the results and discussion of the revised manuscript (pages 22-23, lines 335-344).

p.7 – cavity mutations. Surface mutations of a protein are almost always stable, but mutations in a cavity or protein interior are less likely to be stable. Given that the Gip1 cavity seems to be unstable without something bound, it seems possible that the mutations introduced could impact protein stability or folding. Can the authors comment in the text on what they did to ensure that these were correctly folded?

Our reply: We agree with the reviewer's comment that mutations cause misfolding of proteins. Since we shared this concern, the expression levels and intracellular localization of Trp mutants were observed in this study (Fig. 5b, c). Some Trp mutants, I152W, L187W, L254W, L293W, and L308W, exhibit normal activity with comparable expression levels to other Trp mutants (new Supplementary Fig. 7c). Thus, we cannot see any correlation between stability and activity (page 18, lines 269-272). Of course, these criteria do not fully guarantee normal folding.

Minor comments:

Throughout, "G protein" should not be hyphenated. "G protein-coupled receptor" is

hyphenated correctly in the text.

Our reply: We thank the reviewer for this careful reading. In accordance with this comment, we used “G protein” throughout the revised manuscript.

Introduction, Paragraph 1, last sentence. “The structural basis of GPCR signaling has been extensively studied...” The authors may want to clarify that they mean the basis of GPCR signaling through G proteins. This is a very broad statement and the references cited to support this statement are unusual, consisting of one review and two papers from the Kobilka lab. Recognizing that one cannot reference all of the many papers in the field, the authors may want to focus on reviews and specifically indicate “as reviewed in (refs)” at the end of the sentence. Alternatively, if the authors have specific statements that are supported by the b2AR structures, the salient points should be explicitly indicated in the text.

Our reply: We thank the reviewer for this advice. Accordingly, we have reworded the sentence in the revised manuscript (page 4, line 48).

The acronym for Gip1 is not defined in the main text. It is customary to define acronyms separately in both the abstract and the main text.

Our reply: We thank the reviewer for the comment. We have spelled out Gip1 in the main text of the revised manuscript (page 5, line 63).

The abstract states that the structure of Gip1 is determined, but the text indicates that it is only the C-terminal region of Gip1 that was crystallized. The authors may want to consider correcting the abstract for accuracy.

Our reply: We thank the reviewer for the careful reading. Accordingly, we have corrected the abstract (page 3, line 24).

p.6 second full paragraph starting with “Since Gip1 mainly binds to Gbgsuperscript something”. Whatever is supposed to be in the superscript is garbled.

Our reply: We have corrected this error.

The movies of the crystal structures show a duplication of some of the atoms of the lipid head group.

Our reply: The reviewer is correct. Because the electron density was derived from the mixture of PE and PG shown by a TLC assay (new Supplementary Fig. 2c), the ligands are placed as alternative conformers of phosphatidylethanolamine (PE) and phosphatidylglycerol (PG). A phospholipid was modelled by alternative conformers representing the headgroups of PE and PG with a common glycerophospholipid moiety. For this reason, atoms of the head group are duplicated. A description of the modelling of the lipid ligand has been added to the Methods section of the revised manuscript, and the head group is linked with its glycerophospholipid moiety by bonds in the new Fig. 1b and Supplementary Movie 2 (page 30, lines 448-454).

Reviewers' Comments:

Reviewer #1:

Remarks to the Author:

The authors have addressed my concerns. I support its publication in Nature Comm.

Reviewer #2:

Remarks to the Author:

The authors have now addressed all my concerns and in my opinion the manuscript is ready for publication...